# Upregulated Ca^2+^ Release from the Endoplasmic Reticulum Leads to Impaired Presynaptic Function in Familial Alzheimer’s Disease

**DOI:** 10.3390/cells11142167

**Published:** 2022-07-11

**Authors:** Temitope Adeoye, Syed I. Shah, Angelo Demuro, David A. Rabson, Ghanim Ullah

**Affiliations:** 1Department of Physics, University of South Florida, Tampa, FL 33620, USA; tadeoye@usf.edu (T.A.); syedislamudd@usf.edu (S.I.S.); rabson@usf.edu (D.A.R.); 2Department of Neurobiology and Behavior, University of California, Irvine, CA 92697, USA; ademuro@uci.edu

**Keywords:** neuronal calcium signaling, endoplasmic reticulum, Alzheimer’s, IP_3_R, neurotransmitter release, synaptic transmission, short-term plasticity, facilitation, depression, synchronous release, asynchronous release

## Abstract

Neurotransmitter release from presynaptic terminals is primarily regulated by rapid Ca^2+^ influx through membrane-resident voltage-gated Ca^2+^ channels (VGCCs). Moreover, accumulating evidence indicates that the endoplasmic reticulum (ER) is extensively present in axonal terminals of neurons and plays a modulatory role in synaptic transmission by regulating Ca^2+^ levels. Familial Alzheimer’s disease (FAD) is marked by enhanced Ca^2+^ release from the ER and downregulation of Ca^2+^ buffering proteins. However, the precise consequence of impaired Ca^2+^ signaling within the vicinity of VGCCs (active zone (AZ)) on exocytosis is poorly understood. Here, we perform in silico experiments of intracellular Ca^2+^ signaling and exocytosis in a detailed biophysical model of hippocampal synapses to investigate the effect of aberrant Ca^2+^ signaling on neurotransmitter release in FAD. Our model predicts that enhanced Ca^2+^ release from the ER increases the probability of neurotransmitter release in FAD. Moreover, over very short timescales (30–60 ms), the model exhibits activity-dependent and enhanced short-term plasticity in FAD, indicating neuronal hyperactivity—a hallmark of the disease. Similar to previous observations in AD animal models, our model reveals that during prolonged stimulation (~450 ms), pathological Ca^2+^ signaling increases depression and desynchronization with stimulus, causing affected synapses to operate unreliably. Overall, our work provides direct evidence in support of a crucial role played by altered Ca^2+^ homeostasis mediated by intracellular stores in FAD.

## 1. Introduction

Alzheimer’s disease (AD) is the most common and burdensome of the late-onset degenerative dementias: the World Alzheimer’s Report estimated a global prevalence of over 50 million worldwide, a number expected to triple by 2050 [1,2]. AD manifests as progressive memory impairment initially and a faster rate of cognitive decline and neurodegeneration in later stages, along with behavioral and physiological manifestations [3]. Despite the convoluted etiology of AD, experimental and theoretical investigation suggests that synapses are the primary targets in the early stage of the disease [4]. Histologically, the AD brain is marked by the extracellular deposition of senile beta-amyloid (Aβ) plaques—in early-onset FAD, this accumulation has been traced to abnormalities in the genes encoding amyloid precursor protein (APP) or intramembrane protease presenilin 1 and 2 (PS1, PS2). While rare (5% of AD cases), this form of Alzheimer’s disease holds an estimated heritability over 90% [5]. These abnormalities are accompanied by the intracellular accumulation of neurofibrillary tangles (NFTs)—composed of hyperphosphorylated tau proteins (pTau)—that litter the cerebral and hippocampal cortices [6,7,8,9]. Although the exact mechanism is still being debated, numerous experimental studies implicate elevated intracellular Ca^2+^ levels as one of the main mechanisms underlying Aβ toxicity [10,11]. These studies show that the AD brain is surfeit with dysregulation of Ca^2+^ signaling pathways [10,11,12,13], motivating researchers to propose the Ca^2+^ hypothesis of AD and aging [14]. Indeed, it has been shown that both intra- and extracellular Aβ oligomers and FAD-causing mutations in presenilin result in enhanced Ca^2+^ release from the ER through inositol (1,4,5)-triphosphate (IP3) receptors (IP_3_Rs) and/or ryanodine receptors (RyRs) [10,11,12,13,14,15,16,17,18,19,20,21,22]. This upregulated Ca^2+^ release can contribute to aberrant plasticity and the functional disruption of neuronal networks [11,23].

Intracellular Ca^2+^ is an important second messenger, regulating a multitude of neuronal functions, including neurotransmitter release. Synaptic function at nerve terminals is tightly coupled to the intracellular Ca^2+^ concentration [Ca^2+^], as Ca^2+^ primarily regulates the biological machinery responsible for exocytosis and short-term plasticity [24,25]. The precise temporal control of synaptic transmission by Ca^2+^ is achieved via local signal transduction mechanisms that aim to regulate Ca^2+^ excitability at the axonal bouton. Voltage-gated Ca^2+^ channels (VGCCs) are the primary mediators of the transduction of depolarization-induced Ca^2+^ transients into neurotransmitter release. Furthermore, Ca^2+^ influx through VGCCs leads to physiological events that alter plasma membrane functions underpinning synaptic plasticity, protein expression, spine maintenance, and the regulation of excitability in excitatory synapses [26,27]. Likewise, investigations of the Ca^2+^ dependence of vesicular release have highlighted the role of intracellular stores in Ca^2+^ handling and spontaneous exocytosis [28,29]. Thus, the close association between these Ca^2+^ pathways, their effect on numerous neuronal processes, and their high sensitivity to pathological perturbations make it especially valuable to elucidate the exact nature of the coupling.

Extensive evidence supports the presence of the ER in the nerve terminal of CA3 pyramidal neurons [23,28,29,30,31,32]. In neurons, activation of Ca^2+^-sensitive channels such as IP_3_Rs and RyRs triggers the release of Ca^2+^ from the ER. Opening of IP_3_Rs primarily depends on Ca^2+^ and IP_3_. To achieve this, glutamate released into the synaptic cleft elicits the production of IP_3_ by the activation of membrane-bound metabotropic glutamate receptors (mGLuRs). RyRs activation, on the other hand, is largely controlled by cytosolic [Ca^2+^]. This specialized cascade underscores the importance of IP_3_Rs and RyRs in the regulation of the Ca^2+^-induced Ca^2+^ release (CICR) mechanism of the ER. Previous works have shown that CICR is necessary for ER stores to adequately influence spontaneous vesicle release and homosynaptic plasticity [28,33]. Indeed, in vitro studies confirm that properly sensitized CICR is necessary for normal synaptic function, whereas aberrant CICR underlies the presynaptic impairment associated with AD [34,35,36,37]. Despite this evidence, the precise role of ER Ca^2+^ handling in action potential (AP)-evoked presynaptic Ca^2+^ dynamics and its downstream effect on presynaptic neuronal processes remain unclear [28,34,35,37]. 

Information encoding at the CA3 to CA1 synapses in the hippocampus, which is crucial for learning and memory storage, relies on the spatiotemporal organization of Ca^2+^ events leading up to synaptic transmission [38]. AP arrival at the nerve terminal activates VGCCs, leading to high-amplitude, short-lived Ca^2+^ influx events into the AZ. Coupled with this specialized pathway, Ca^2+^ sensors initiate a heterogeneous fusion of neurotransmitter vesicles with the plasma membrane that often culminates in either fast synchronous or slow asynchronous release. Neuronal communication primarily relies on the synchronous mode of exocytosis, which is regulated by synaptotagmin-1 (Syt1) sensors with low Ca^2+^ affinity and rapid kinetics that are critical for the exquisite temporal precision of vesicle fusion that characterizes synaptic transmission at most CA3-CA1 terminals [39,40,41]. Such a high degree of synchrony is in part achieved by the steep dose dependence of evoked release on the short-lived Ca^2+^ transients constrained to micro- or nanodomains within the vicinity of VGCCs [24,42]. A global buildup of [Ca^2+^], on the other hand, accelerates the recruitment of release-ready vesicles, controlling the degree of synaptic plasticity [43]. Thus, changes in Ca^2+^ signals at the local or global scale are expected to disrupt synaptic transmission and plasticity. Consistent with this assertion, experimental manipulations that perturb evoked Ca^2+^ influx alter the contribution of the synchronous mode of release to overall exocytosis and compromise synaptic plasticity [44,45,46]. These findings highlight the need for a thorough investigation of the potential link between impaired synaptic function and disrupted Ca^2+^ homeostasis in AD.

In this study, we incorporate findings from extensive experimental and computational studies to develop a detailed biophysical model of Ca^2+^-driven exocytosis at the CA3 presynaptic terminal. The model accounts for the observed Ca^2+^ and IP_3_ signaling pathways necessary for intracellular Ca^2+^ regulation and integrates the elaborate kinetics of neurotransmitter release—vesicle docking, mobilization, priming, and fusion—aided by distinct Ca^2+^ sensors. We reproduce crucial statistics of both Ca^2+^ and release events reported at small excitatory synapses, such as transient timescale, amplitude, and decay time. By developing a mathematical framework for coupling the Ca^2+^ domains surrounding the ER and AZ, we study how FAD-associated pathological Ca^2+^ release from the ER disrupts presynaptic neurotransmitter release rates and consequently alters synaptic plasticity and facilitation at affected synapses. Overall, our work provides novel insights on the pathologic role of aberrant neuronal Ca^2+^ handling on glutamate release and the downstream effects on synaptic dysfunction and cognitive decline observed in FAD. We address the limitations of this model, noting that the inclusion of the differential enrichment of VGCC subtypes, mitochondrial function, Ca^2+^ buffers, and RyRs activity in future extensions of the current work will provide a comprehensive computational framework that can be used to investigate key cellular mechanisms and processes, which in turn can be targeted for reversing presynaptic impairment in FAD.

## 2. Materials and Methods

### 2.1. Calcium Model

Building on the extensive literature, we capture intracellular Ca^2+^ dynamics by first developing a compartmental model of a hippocampal CA3 axonal bouton, which includes the main fluxes that invade the bulk cytosol as well as regulatory mechanisms present in the ER [47,48,49,50] (Figure 1). Our canonical synaptic bouton is modelled as a sphere with fixed volume $V_{bouton}$ = 0.122 μm^3^, in agreement with findings from the ultrastructural analysis of hippocampal synapses [51,52]. We consider an average of 1.3 AZs in small hippocampal boutons implemented in a spherical AZ (with area = 0.04 μm^2^) [51,52]. Although we assume a well-mixed cytoplasm, we next incorporate two microdomains of sharp Ca^2+^ transients produced by clusters of IP_3_Rs and VGCCs proximal to the ER and plasma membrane, respectively. Because the VGCCs (P/Q-type Cav2.1 channels) implemented here are spatially distributed in small clusters within the AZ, we implement a characteristic 25 nm cluster [52,53]. To account for the spatial extent of the Ca^2+^ domain in the vicinity of the VGCC cluster, we use the findings in cortical pyramidal terminals that show that low mM concentrations of the slow Ca^2+^ buffer ethylene glycol tetra-acetic acid potently attenuate transmitter release [27,54,55] for guidance. We consider a cytosol-to-VGCC microdomain ratio of 60, assuming a domain of elevated [Ca^2+^] that extends over more than 100 nm [27,56]. As a result of these considerations, the Ca^2+^ dynamics in the respective compartments as well as the entire bouton ([$Ca_{total}^{2+}$]) is described by four coupled non-linear ODEs (Equations (1)–(4)). Table 1 defines the fluxes (J) in terms of various Ca^2+^ concentrations, along with volume fractions. The parameters used are listed in Table 2 and Table 3. IP_3_ is a critical second messenger to Ca^2+^ The pathways for its metabolism are succinctly described in Equation (5) [57,58], with further details in Table 4. Inhomogeneity of ligands persists throughout the bouton; however, we assume spatially homogenous compartments, and only track the temporal evolution of ligands.
(1)$$\frac{d}{dt}\left[Ca_{cyt}^{2+}\right]=J_{in}+J_{IPR-diff}-J_{PMCA}+J_{ER-leak}+J_{VGCC-diff}-J_{SERCA}$$
(2)$$\frac{d}{dt}\left[Ca_{IPR_{n}}^{2+}\right]=\delta_{1}\left(J_{IPR}-J_{IPR-diff}\right)+J_{coupling}$$
(3)$$\frac{d}{dt}\left[Ca_{AZ}^{2+}\right]=\delta_{3}\left(J_{VGCC}-J_{VGCC-diff}\right)-\frac{1}{\delta_{1}}J_{coupling}$$
(4)$$\frac{d}{dt}\left[Ca_{total}^{2+}\right]=J_{in}-J_{PMCA}+J_{VGCC}$$
(5)$$\frac{d}{dt}\left[IP_{3}\right]=\frac{1}{\tau_{IP_{3}}}\left(J_{PLC}-J_{deg}\right)$$
where $\left[Ca_{cyt}^{2+}\right]$ is the Ca^2+^ concentration in the cytosol, $\left[Ca_{IPR_{n}}^{2+}\right]$ is the Ca^2+^ concentration in the microdomain surrounding the IP_3_Rs, and $\left[Ca_{AZ}^{2+}\right]$ represents the Ca^2+^ concentration surrounding the small cluster of VGCCs in the AZ.

The Ca^2+^ concentration in the ER is given by $\left[Ca_{ER}^{2+}\right]=\delta_{2}\left(Ca_{total}^{2+}-Ca_{cyt}^{2+}+Ca_{IPR_{n}}^{2+}/\delta_{1}-Ca_{AZ}^{2+}\right)$. $\delta_{1}$, $\delta_{2}$, and $\delta_{3}$ represent the volume ratios of the intracellular compartments and are explained in Table 3. Fluxes in our model were selected to account for the essential regulating components of intracellular Ca^2+^ signaling. $J_{in}$ represents the Ca^2+^ entry through plasma membrane channels, such as store-operated Ca^2+^ channels (SOCC) and basal plasma membrane leak. $J_{IPR}$ represents release from the ER through IP_3_Rs, whereas Ca^2+^ diffusion from the microdomain around IP_3_R clusters to the bulk cytosol is modelled by $J_{IPR-diff}$. Likewise, $J_{VGCC-diff}$ and $J_{VGCC}$ are included to account for Ca^2+^ diffusion from the AZ to the bulk cytoplasm and influx through VGCCs, respectively. Ca^2+^ efflux from the intracellular compartment by plasma membrane Ca^2+^ ATPase (PMCA) is captured by $J_{PMCA}$, and $J_{ER-leak}$ is the Ca^2+^ leak from the ER. Sequestering of Ca^2+^ from the cytoplasm into the ER through Sarco/Endoplasmic Reticulum Ca^2+^ ATPase (SERCA) is represented by $J_{SERCA}$.

To investigate the effect of altered ER Ca^2+^ handling on vesicular fusion situated in the AZ, we incorporate a flux ($J_{coupling}$) intended to mimic the close association of the ER with the nerve terminal of CA3 pyramidal neurons [30,31,32]. Based on evidence of the existence of a feedback loop between synaptic function and ER Ca^2+^ content, we build a bidirectional model of Ca^2+^ exchange. We assume the simple transfer of Ca^2+^ between the two microdomains, potentially mediated by Ca^2+^ buffering and enzymatic proteins. This coupling is modelled by an equation analogous to bidirectional models of SERCA flux (Equation (6)) [50], where $V_{c}$ is the maximum flux from the AZ to the microdomain around IP_3_R cluster and *K_c_* determines the half-maximal transfer rate.
(6)$$J_{coupling}=V_{c}\frac{{\left[Ca_{AZ}^{2+}\right]}^{2}-\bar{k}\left({\left[Ca_{IPR_{n}}^{2+}\right]}^{2}\right)}{{\left[Ca_{AZ}^{2+}\right]}^{2}-K_{c}{}^{2}}$$

There is strong evidence that the expression of Ca^2+^ buffering proteins in AD-affected neurons is significantly lower than in WT neurons [59,60,61]. We incorporate these observations into our model by using parameter values that result in a stronger coupling in the AD-affected bouton as compared to the WT bouton. Although the choice of the model has its limitations, using a model that captures buffered Ca^2+^ diffusion becomes numerically intractable quickly as distances approach the physiologically reasonable order of nanometers. Thus, this approach provides an extremely useful method to account for the interplay of [$Ca_{AZ}^{2+}$] and [$Ca_{IPR_{n}}^{2+}$]. 

### 2.2. IP_3_R Model

In the past, several models for IP_3_Rs have been developed in WT and AD-affected cells [18,62,63,64]. All these models are based on data obtained from non-neuronal cells. While these models can be used to make reliable qualitative predictions, our goal here is to quantify the effect of upregulated Ca^2+^ signaling on neurotransmitter release, where a small difference in the open probability of the channel or dwell times can result in a significant change due to the small volume of the synaptic terminal. Thus, we developed a new four-state model to implement the kinetics of IP_3_Rs (Figure 2A). The channel has zero, two, two, and five Ca^2+^ bound when in the Resting (R), Active (A), Open (O), and Inactive (I) states, respectively. The transition rates between different states and the corresponding parameters are reported in Table 5 and Table 6, respectively. As shown in Figure 3, the model closely fits the kinetics of IP3R in neurons from WT and 3xTg AD mice observed in [21]. In the cluster, the gating of each IP_3_R is regulated by the Ca^2+^ concentration in the microdomain around the cluster.

### 2.3. Membrane Voltage Dynamics

The basic equations for the membrane potential used in our model are adopted from Ref. [65]. Membrane potential (*V*) is governed by primary Na^+^ ($I_{Na}$), K^+^ ($I_{K}$), Cl^−^ ($I_{Cl}$), and Ca^2+^ ($I_{Ca}$), currents, as well as the current due to applied stimulation ($I_{app}$), and is given as
$$C_{m}\frac{dV}{dt}=I_{app}+I_{Na}+I_{K}+I_{Cl}+\left(\frac{I_{Ca}}{Area}\right)$$
where we assume standard specific membrane capacitance $C_{m}$ of $1\frac{\mu F}{cm^{2}}$. The Na^+^ and K^+^ concentrations are assumed to be fixed, with corresponding currents consisting of active and passive leak components given by
$$I_{Na}=-\left(g_{Na}m_{\infty }{}^{3}h\right)\left(V-E_{Na}\right)-g_{Na_{leak}}\left(V-E_{Na}\right)$$
$$I_{K}=-\left(g_{K}n^{4}+\frac{g_{AHP}{\left[Ca^{2+}\right]}_{cyt}}{1+{\left[Ca^{2+}\right]}_{cyt}}\right)\left(V-E_{K}\right)-g_{K_{leak}}\left(V-E_{K}\right)$$

Chloride currents only consist of the passive leak contribution, defined by
$$I_{Cl}=-g_{Cl_{leak}}\left(V-E_{cl}\right)$$


The steady state gating activation and inactivation variables, as well as associated channel forward and reverse rates, are calculated using the equations in Table 7. Various parameters used in the membrane potential equations are listed in Table 8. 

**Table 7 cells-11-02167-t007:** Parameter values for membrane potential equations.

Parameter	Value/Units [65]	Description
$C_{m}$	1 μF/cm^2^	Membrane capacitance
$g_{Na}$	120 mS/cm^2^	Maximum conductance for active Na^+^ channels
$g_{Na_{leak}}$	0.0175 mS/cm^2^	Conductance for Na^+^ passive leak channels
$g_{K}$	36 mS/cm^2^	Maximum conductance for active K^+^ channels
$g_{K_{leak}}$	0.05 mS/cm^2^	Conductance for K^+^ passive leak channels
$g_{Cl_{leak}}$	0.05 mS/cm^2^	Conductance for Cl^−^ passive leak channels
ϕ	5.0	
$g_{AHP}$	0.01	
$\alpha_{n}$	$\frac{0.01\left(V+34\right)}{1-\exp\left(-\frac{V+34}{10}\right)}$	Forward rate for K^+^ current activation gating variable
$\beta_{n}$	$0.125\exp\left(-\frac{V+44}{80}\right)$	Backward rate for K^+^ current activation gating variable
$\alpha_{h}$	$0.07\exp\left(-\frac{V+44}{20}\right)$	Forward rate for Na^+^ current inactivation gating variable
$\beta_{h}$	$\frac{1}{1+\exp\left(-\frac{V+14}{10}\right)}$	Backward rate for Na^+^ current inactivation gating variable
$\alpha_{m}$	$\frac{0.1\left(V+30\right)}{1-\exp\left(-\frac{V+30}{10}\right)}$	Forward rate for Na^+^ current activation gating variable
$\beta_{m}$	$4\exp\left(-\frac{V+55}{18}\right)$	Backward rate for Na^+^ current activation gating variable
$m_{\infty }$	$\frac{\alpha_{m}}{\alpha_{m}+\beta_{m}}$	Occupancy of Na^+^ activation gating variable
	$\frac{dn}{dt}=\phi \left(\alpha_{n}\left(1-n\right)-\beta_{n}n\right)$	Evolution occupancy of gating K^+^ current gating variable
	$\frac{dh}{dt}=\phi \left(\alpha_{h}\left(1-h\right)-\beta_{h}h\right)$	Evolution occupancy of gating Na^+^ current inactivation gating variable

**Table 8 cells-11-02167-t008:** Reaction rates for kinetic schemes for VGCC.

Parameter	Value/Units	Notes
$\alpha_{10}$ $,\;\alpha_{20}$ $,\;\alpha_{30}$ $,\;\alpha_{40}$	4.04, 6.70, 4.39, 17.33 ms^−1^	From [66,67]
$\beta_{10}$ $,\;\beta_{20}$ $,\;\beta_{30}$ $,\;\beta_{40}$	2.88, 6.30, 8.16, 1.84 ms^−1^	From [66,67]
$k_{1}$ $,\;k_{2}$ $,\;k_{3}$ $,\;k_{4}$	49.14, 42.08, 55.31, 26.55 mV	From [66,67]

**Table 9 cells-11-02167-t009:** Reaction rates for kinetic schemes for exocytosis.

Parameter	Value/Units	Description	Notes
$k_{mob}$	Mobilization rate	5.0 × 10^−5^ μM^−1^ms^−1^	From [68]
$k_{demob}$	Demobilization rate	0.0022 ms^−1^	From [68]
$k_{priming\;}$	Priming rate	0.027990 μM^−1^ms^−1^	From [68]
$k_{unpr\;}$	Unpriming rate	0.005356 ms^−1^	From [68]
$k_{attach\;}$	Attachment rate	0.0015 μM^−1^ms^−1^	Fit
$k_{detach\;}$	Detachment rate	0.001158 ms^−1^	Fit
$k_{RF\;}$	Refractoriness	10.34–1 ms^−1^	Fit
α	Association rate for synchronous release	0.061200 μM^−1^ms^−1^	From [67]
β	Dissociation rate for synchronous release	2.32 ms^−1^	From [67]
λ	Association rate for asynchronous release	0.002933 μM^−1^ms^−1^	Fit
δ	Dissociation rate for asynchronous release	0.014829 ms^−1^	Fit
$\gamma_{1}$	Spontaneous release rate	9 × 10^−6^ ms^−1^	Fit
$\gamma_{2}$	Synchronous release rate	2.000008 ms^−1^	From [67]
$\gamma_{3}$	Asynchronous release rate	$a\cdot \gamma_{2}$ ms^−1^	From [67]
b	Cooperativity factor	0.250007	From [67]
a		0.025007	From [67]

**Figure 2 cells-11-02167-f002:**
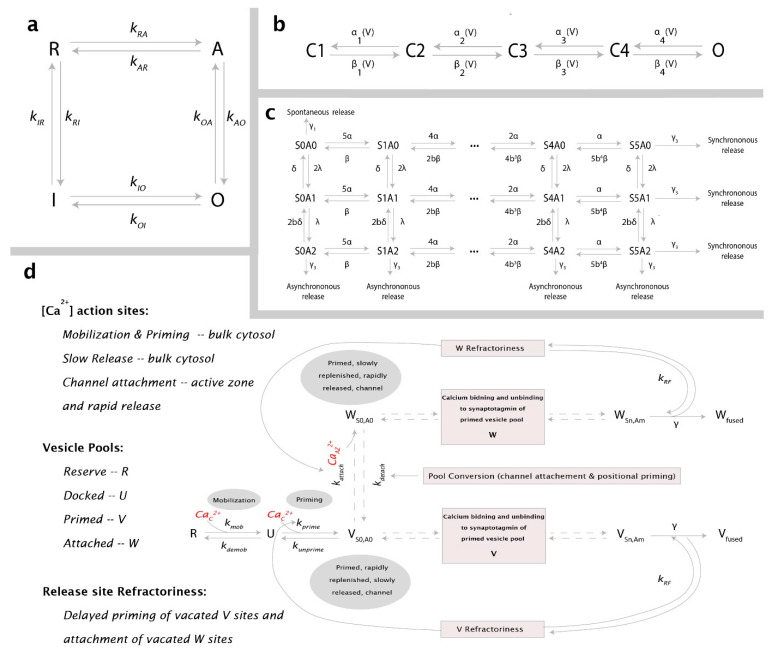
Kinetic schemes used in the model: (**a**) Gating kinetics of IP_3_R. Four−state model representing the possible states along with corresponding transition rates; (**b**) Model for VGCC gating with four closed states (C1–C4) and one open Ca^2+^ conducting state (O); (**c**) Scheme for Ca^2+^ binding to synaptotagmin with dual Ca^2+^ sensors for fast synchronous (S with five Ca^2+^-binding sites), slow asynchronous (A with two Ca^2+^−binding sites), and spontaneous exocytosis; (**d**) Adapted with permission from Ref. [68]. 2009, Elsevier Inc. The overall release scheme, which includes vesicle mobilization from a reserve pool (R) to docked, unprimed pool (U), molecular priming to vesicles unattached to a Ca^2+^ channel (V), and conversion to vesicles coupled to a VGCC cluster (W). Both vesicle pools are released through the dual sensor release model. Channel−attached vesicles are strongly dependent on [$Ca_{AZ}^{2+}$], whereas [$Ca_{cyt}^{2+}$ ] governs the release of detached vesicles. Reaction rates along with respective references are listed in Table 6, Table 8, and Table 9.

**Figure 3 cells-11-02167-f003:**
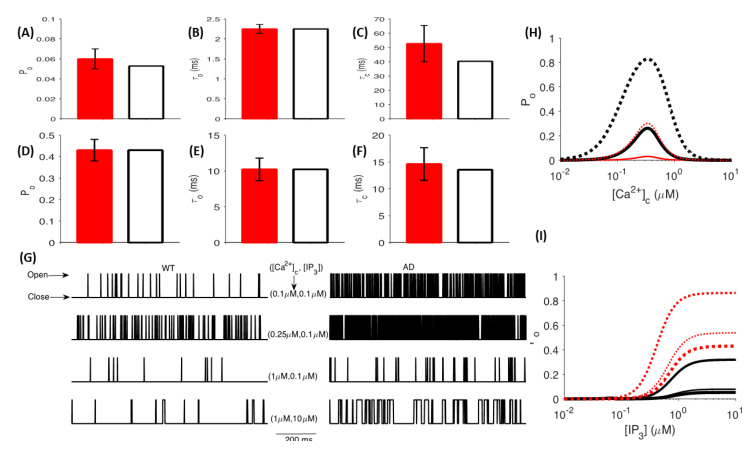
Gain−of−function enhancement of IP_3_R gating in FAD: The *P_o_* (**A**,**D**), *τ_o_* (**B**,**E**), and *τ_c_* (**C**,**F**) of IP_3_R given by the model (empty bars) and observed values in primary cortical neurons from WT (**A**–**C**) and 3xTg AD mice (filled bars) (**D**–**F**) at [$Ca_{cyt}^{2+}$] = 1 μM and [$IP_{3}$ ] = 10 μM; (**G**) Sample time−traces generated by stochastically simulating a single IP_3_R channel in cortical neurons from WT (left column) and AD (right column) mice at different [$Ca_{cyt}^{2+}$ ] and [$IP_{3}$ ] values shown in the figure; (**H**) *P_o_* of IP_3_R as a function of [$Ca_{cyt}^{2+}$ ] at [$IP_{3}$ ] = 0.3 μM (thin lines) and 1 μM (thick lines) in WT (solid lines) and AD-affected (dotted) neurons; (**I**) *P_o_* of IP_3_R as a function of [$IP_{3}$ ] at [$Ca_{cyt}^{2+}$ ] = 0.1, 0.25, and 1 μM (the increasing value of [$Ca_{cyt}^{2+}$ ] is represented by the thickness of the line) in WT (solid lines) and AD-affected (dotted) neurons. Experimental values shown for comparison in (**A**–**F**) are from [21].

### 2.4. Voltage-Gated Ca^2+^ Channels

Consistent with findings in [66], we implement only the predominant high-threshold Cav2.1 (P/Q-type) channels present at presynaptic nerve terminals using a five-state kinetic scheme (see Figure 2b). Voltage-dependent activation and deactivation rates for each closed state (*I =* 1, 2, 3, 4) were, respectively, calculated as follows: $\alpha_{i}\left(V\right)=\alpha_{i0}\exp\left(V/k_{i}\right)$, $\beta_{i}\left(V\right)=\beta_{i0}\exp\left(V/k_{i}\right)$, where values for activation and deactivation rates at 0 mV, $\alpha_{i0}$ and $\beta_{i0}$, and for slope factor $k_{i}$ were taken from [66] and are listed in Table 8. As with the IP_3_Rs, we model VGCC gating stochastically as a discrete-time Markov Chain (DTMC) (*see*
Section 2.7
*Numerical Methods*). Single-channel Ca^2+^ currents are calculated using $I_{s}=gP_{o}\left(V-E_{Ca^{2+}}\right)$, where values for conductance g, extracellular Ca^2+^ concentration, and Nernst potential $E_{Ca^{2+}}$ were obtained from [53] and are reported in Table 8.

### 2.5. Overall Release Model

The complete release scheme has been adopted and modified from [68] (see Figure 2c,d). In addition to the Ca^2+^-dependent vesicle mobilization and priming steps, we replace the independence of Ca^2+^ binding to the C2A and C2B domains of synaptotagmin with the dual sensor model proposed in [69], where two independent Ca^2+^ sensors act in parallel to trigger distinct pathways of exocytosis that lead to fast synchronous, slow asynchronous, and spontaneous release. Synchronous release is mediated by the sensor S with five Ca^2+^-binding sites and cooperativity (b) incorporated to progressively decrease backward rates—Ca^2+^ unbinding. Synchronous fusion occurs when all five binding sites are occupied. Likewise, the sensor A—with two Ca^2+^-binding sites—mediates asynchronous release with the same cooperativity parameter b. Spontaneous release is also included and occurs at a much slower rate when the sensors have no Ca^2+^ bound. As in [67], release rates and model parameters were obtained according to fits to experimental data reported in [70]. Contrary to the dual sensor model of [71], we do not assume the synchronous ($\gamma_{2}$) and asynchronous ($\gamma_{3}$) release rates to be the same. This is because, according to [67], hippocampal release rates from [70] could not be fitted otherwise. Consequently, we use the release rate for the asynchronous release to be aγ, where *a* = 0.025.

As described in [69] and shown in Figure 4, compared to the allosteric model, this model captures the expected heterogeneity and latency of exocytosis more accurately at the Calyx of Held. Apart from the intrinsic heterogeneity of release pathways, the model implemented here captures the heterogeneity of vesicle pools—slow and fast—where docking and priming are part of the upstream processes for recruiting vesicles into the Slow-Releasing Pool (SRP), and super-priming of vesicles in the SRP aids the conversion of SRP vesicles to those in the Fast-Releasing Pool (FRP). The recruitment of vesicles into the SRP is dependent on the cytosolic [$Ca_{cyt}^{2+}$], whereas channel attachment is aided by Ca^2+^ influx. As described in Figure 2, the target of Ca^2+^ mediating vesicle fusion depends on which pool the fusing vesicle belongs to; for vesicles in the FRP, this is $\left[Ca_{AZ}^{2+}\right]$, while those in the SRP bind [$Ca_{cyt}^{2+}$].

We also include release site refractoriness introduced in [67,68] in order to simulate experimental observations. In this context, a vesicle cannot be released from a vacated site for a period determined by $k_{RF}$, such that sites—either Ca^2+^-attached (*W*) or detached (*V*)—from which vesicles had been released remain unable to accept a new primed vesicle for some time. Phasic synapses are known to have briefer refractoriness compared to their tonic counterparts, and, as such, we choose $k_{RF}$ to be 0.01 ms^−1^, similar to values observed at hippocampal synapses in [73]. Different parameters used in the release model are listed in Table 9.

### 2.6. Synchrony Measure

For a wide range of synaptic configurations with distinct intrinsic release probabilities, we computed the synchrony of AP arrival times (estimated at peak) and release event times using a modified version of the Pinsky–Rinzel measure of synchrony [74]. We transformed the firing times T(k) for every *k^th^* event of the neuron into a set of corresponding vector phases ϕ(k) using Equation (7), where $T_{AP}\left(k\right)$ corresponds to all the AP events within the duration of simulation. For each vector phase ϕ(k), we compute the synchrony r(ϕ(k))—numbers between 0 and 1—using the complex order parameter defined by Strogatz and Mirollo averaged across all events for a single synaptic configuration [75] (Equation (8)).
(7)$$\phi \left(k\right)=\frac{T_{AP}\left(k\right)-T\left(k\right)}{T\left(k+1\right)-T\left(k\right)}$$
(8)$$r\left(\phi \left(k\right)\right)=\frac{1}{N_{K}}\sum^{\;}e^{\left(2\pi i\phi \left(k\right)\right)}$$

### 2.7. Numerical Methods

Deterministic equations (Equations (1)–(4)) are solved using the fourth-order Runge–Kutta algorithm (RK4) with a 1 μs time step, while the stochastic states of the IP_3_Rs and VGCCs determined by the corresponding kinetic schemes were simulated using a procedure outlined in [76], which is equivalent to the Gillespie algorithm with fixed time step. All numerical simulations were performed in MATLAB (The MathWorks, Natick, MA, USA) and data analysis was carried out using custom Python scripts (version 3.9).

## 3. Results

### 3.1. The Gain-of-Function Enhancement of IP_3_R Gating in FAD

The exaggerated Ca^2+^ release observed in FAD-affected neurons is ascribed mainly to the gain-of-function enhancement of IP_3_Rs in the affected cells due to FAD-causing mutations in presenilin [18,19,21,77,78]. Indeed, it has been shown that cell models expressing PS mutations exhibit a several-fold increase in the open probability (Po) of IP_3_Rs [19,77]. Specifically, the Po of IP_3_R in cortical neurons from 3xTg-AD mice carrying PS mutations was enhanced by 700% relative to control mice with wildtype (WT) PS (0.43 ± 0.05 in AD versus 0.06 ± 0.01 in WT mice) at [$Ca_{cyt}^{2+}$] of 1 μM and IP_3_ concentration ([$IP_{3}$]) of 10 μM [21]. Given that all models for IP_3_R in WT or AD-affected cells are based on non-neuronal cells, we use the above findings to build a new model for the gating kinetics of IP_3_Rs in neurons from WT and 3xTg-AD mice (see details in Section 2). Parameters yielding the best fit to experimental observations are listed in Table 6.

Our model mimics the gating of IP_3_R in neurons from WT and 3xTg-AD mice (Figure 3), closely reproducing the observed values of the *P_o_* (Figure 3A,D), mean open time (*τ*_o_) (Figure 3B,E), and mean close time (*τ*_c_) (Figure 3C,F) reported in [21]. The significantly higher *P_o_* of the channel in FAD-affected neurons is reflected in the time-traces from the model, showing that the channel spends significantly more time in the open state than in the diseased state (Figure 3G). To determine how the observations about different resting [$Ca_{cyt}^{2+}$] in AD-affected cells change the behavior of IP_3_Rs, we plot the *P_o_* of the channel as a function of [Ca^2+^]_c_ and [$IP_{3}$] (Figure 3H,I). Previous studies of 3xTg and APPSW AD mice models reported resting [$Ca_{cyt}^{2+}$] of 247 ± 10.1 nM and 225.2 ± 11.7 nM, respectively, whereas [${Ca}_{cyt}^{2+}$] of 110.8 ± 1.5 nM was recorded in WT mice [79]. In particular, cortical neurites of plaque-bearing mice express a six-fold increase in resting [$Ca_{cyt}^{2+}$] relative to non-transgenic mice [11]. Here, our model exhibits a 4.42-fold increase in the *P_o_* of IP_3_Rs in WT neurons as we increase [$Ca_{cyt}^{2+}$] from 110 nM to 250 nM (0.005626 vs. 0.02484) at 0.3 μM [$IP_{3}$]. At [$IP_{3}$] = 0.3 μM and [$Ca_{cyt}^{2+}$] = 250 nM, the *P_o_* of the channel in FAD-affected neurons reaches 0.2565—a 10.32-fold increase relative to WT neurons (Figure 3H,I). Thus, our model predicts that an IP_3_R in FAD-affected neurons will exhibit an almost 45-fold increase in *P_o_* compared to control neurons with the same amount of IP_3_, leading to significantly higher Ca^2+^ release from the ER. 

### 3.2. Characterization of the Glutamate Release Model and Release Event

We next examine the relationship between total release from a single process following [$Ca_{cyt}^{2+}$] clamps at different concentration steps. We observe that the release rate rapidly increases and transiently decays back to basal levels within tens of milliseconds—a result of the sensitivity of the Ca^2+^ sensors (Figure 4A). For comparison, we also show the release rate given by the allosteric model at different [$Ca_{cyt}^{2+}$] values (Figure 4B). Moreover, we obtain results similar to quantitative studies of transmission profiles in the Calyx of Held synapse, where release profiles were quantified from the deconvolution of evoked miniature excitatory postsynaptic current waveforms [69]. Unsurprisingly, we observed a shift in the [$Ca_{cyt}^{2+}$] dependency of the peak release rate as the response to clamped intracellular Ca^2+^ levels is both lower and right-shifted relative to the experimental data for the Calyx of Held (Figure 4C), whereas the time delay (time-to-peak) of the peak release rate shows a higher and right-shifted exponential decay (Figure 4D). This is consistent with observations of approximately a hundred-fold decrease in total vesicle population in hippocampal boutons [51,80]. In addition, our model mimics the Ca^2+^-dependent increase in release rate observed at high-fidelity synapses of cerebellar mossy fiber boutons (cMFBs), which permit direct presynaptic recordings and are reported to have high structural similarities with their hippocampal counterparts (Figure 4C) [72,81]. Moreover, in agreement with previous recordings of spontaneous release events from CA3-CA1 synapses, we affirmed that the spontaneous release rate elicited by resting level [$Ca_{cyt}^{2+}$] of 100 nM is within the reported range of 10^−4^ and 10^−5^ per ms [53,82,83]. The dual sensor model is more in line with the findings on CA3-CA1 synapses and is used throughout this work. However, it should be mentioned that some technical details may prevent direct comparison to exocytosis at cMFBs where the release rate is obtained as the inverse of exponential fits to the cumulative release obtained from deconvolution analysis and reflects the release rate constant per vesicle. Therefore, the comparison between our model and these findings must be viewed cautiously [69,72].

### 3.3. FAD-Associated Intracellular Ca^2+^ Changes Enhance Neurotransmitter Release

Several studies have reported enhanced Ca^2+^ release from the ER in AD-affected neurons [84,85,86,87,88]. This enhanced Ca^2+^ release has been associated with the several-fold increase in the *P_o_* of IP_3_Rs observed in multiple animal and human cell models of FAD [18,19,21,79,85]. Although presynaptic plasticity and synaptic vesicle release (SVR) are tightly coupled to Ca^2+^ entry through VGCCs, several studies have established an important role for ER stores in regulating presynaptic plasticity and neurotransmission [28,34,35,89,90]. Furthermore, strong experimental evidence supports the existence of a feedback loop between the ER Ca^2+^ stores and AP-triggered exocytosis events [91]. Accordingly, here, we explore how the observed gain-of-function enhancement of IP_3_Rs in FAD affects neurotransmitter release. In addition to enhanced Ca^2+^ release through IP_3_Rs, multiple studies have implicated a significant downregulation of Ca^2+^ buffering proteins in FAD-affected neurons as compared to WT neurons [59,60,61]. We incorporate the effect of observed changes in Ca^2+^ buffering proteins by considering two configurations: High Coupling (HC) and Normal Coupling (NC) between the ER and AZ. The HC configuration corresponds to the downregulation of Ca^2+^ buffering proteins (see more details in Section 2) (Appendix A).

In Figure 5A, we show a typical release profile in response to a single AP, where a clear difference between WT and FAD-affected synapses can be seen. To quantify this difference, we compute the release probability (P_r_) by counting the number of vesicles released from the slow and fast release-ready pools (RRP), divided by the number of vesicles initially in both pools. In the FAD-affected synapse, we observe enhanced P_r_ over a wide range of VGCC expression in a dose–response manner (Figure 5B), suggesting that the acute effects of FAD-driven aberrant cytosolic Ca^2+^ are not exclusive to synapses operating in the regime of sparse VGCCs. Surprisingly, we observe only a marginal difference in the peak release rate between WT and FAD-affected synapses (Figure 5C). We also calculated the number of vesicles released during a single AP by integrating the rate of vesicle release from the slow and fast RRP and noticed a significant increase in vesicles released in the FAD-affected synapse (Figure 5D). To discern the contribution of enhanced Ca^2+^ release through IP_3_Rs from that due to the HC, we simulate four scenarios: (1) enhanced Ca^2+^ release through IP_3_Rs but NC (AD-NC), (2) enhanced Ca^2+^ release through IP_3_Rs with HC (AD-HC), (3) normal Ca^2+^ release through IP_3_Rs and NC (WT-NC), and (4) normal Ca^2+^ release through IP_3_Rs and HC (WT-HC). We notice that while HC causes a minor increase in the release probability and vesicles released in the WT synapse, it strongly affects both these features in the FAD-affected synapse (Appendix A).

To gain deeper insights into the observed changes in the P_r_, we examined the dependence of the release rise time (time to peak release rate) on P_r_ (P_r_ is increased by increasing the number of VGCCs, as in Figure 5A) and observed no significant differences between FAD-affected and WT synapses (Figure 5E). Interestingly, the decay time (time to basal release rate) exhibits a biphasic dependence on P_r_, with a longer decay time in FAD-affected synapses. Strikingly, the decay time as well as the concomitant FAD-associated enhancement is attenuated in synapses with both high and low P_r_, and peaks at P_r_ corresponding to physiologically reasonable VGCCs expression for small hippocampal synapses [84], suggesting that such small hippocampal synapses are more sensitive to alterations due to FAD-associated Ca^2+^ disruptions (Figure 5F). Similar to the release probability as a function of time and vesicles released during a single AP, HC has a stronger effect on the release rise time, mainly due to the changes in Ca^2+^ in AZ in the FAD-affected synapse (Appendix A).

To assess the direct correspondence between the changes in different aspects of SVR and enhanced Ca^2+^ release, we examine the cumulative Ca^2+^ at AZ. The elevated Ca^2+^ release from the ER maintains larger cumulative Ca^2+^ (area under the Ca^2+^ transient) at AZ in the FAD-affected synapse consistently across a wide range of VGCC expression, in agreement with several studies showing that ER Ca^2+^ channels can sculpt the spatiotemporal dynamics of exocytosis and consequently neuronal function (Figure 5G) [92,93]. However, this continuous enhancement as a function of VGCC expression is not consistent with the biphasic behavior of the time to basal rate as a function of P_r_. Next, we examined the residual Ca^2+^ in the AZ, obtained as the cumulative Ca^2+^ that persists during the decay phase of the Ca^2+^ transient. Our results show that larger residual Ca^2+^ in FAD-affected synapses also exhibits a biphasic behavior as a function of P_r_, similar to the time to basal neurotransmitter release rate. Again, this enhancement was non-uniform, suggesting that small hippocampal synapses with intermediate P_r_ values are highly sensitive to pathological alterations (Figure 5H). Taken together, our data reveal that ER-driven Ca^2+^ disruption plays a critical role in shaping the observed response profile, with the acute effects induced by such disruptions more severely expressed in small hippocampal synapses. Our results also show that these effects are more sensitive to the coupling between the Ca^2+^ domains in the vicinity of VGCCs and ER in FAD-affected synapses (Appendix A), consistent with reports on the involvement of ER Ca^2+^ in the regulation of presynaptic resting [Ca^2+^]_c_ and neurotransmission [91,92,93]. Furthermore, the sensitivity to the coupling between the Ca^2+^ domains is exacerbated for synapses with intermediate P_r_ values (Appendix A). 

### 3.4. Very Short-Term Plasticity Is Enhanced in the FAD-Affected Synapse

Next, we investigated how the enhanced Ca^2+^ release from the ER affects very short-term presynaptic plasticity (STP). STP is assessed by determining the paired-pulse ratio (PPR): a classical measure of presynaptic modulation in response to paired stimuli separated by very short time intervals [73]. After stimulating the nerve terminal with two pulses separated by a 40 ms interval (Figure 6A), we define PPR as the ratio of the P_r_ following the second pulse ($P_{r2}$) to that of the first pulse ($P_{r1})$ averaged over several trials. Therefore, the response to the second stimulus can either be enhanced with $P_{r2}/P_{r}$ > 1 (short-term facilitation (STF)) or depressed with $P_{r2}/P_{r}$ < 1 (short-term depression (STD)).

On average, both WT and FAD-affected synapses exhibited an inverse relationship between $P_{r1}$ and the PPR, consistent with previous findings in phasic synapses, such as small glutamatergic synapses in the hippocampus (Figure 6B) [93,94,95]. This negative correlation, thought to be a universal feature of these synapses, is assumed to be caused by the spike-driven depletion of vesicles in the RRP after the first pulse, which is unlikely to be recovered by Ca^2+^-driven facilitation upon the next stimulation. Therefore, whether a synapse exhibits STF or STD is largely dependent on the recent activation history, which implies that synapses with a large number of VGCCs and consequently very high intrinsic release probabilities tend to depress their response more severely to a second pulse, allowing them to operate at low PPR [25,73,93,94,95,96].

While additional mechanisms may contribute to STD, the depletion model of depression in phasic synapses suggests that, at rest, the priming sites containing the RRP of vesicles are mostly occupied, which reflects the inability of residual Ca^2+^—left over from the previous stimulation—and the incoming flux to potentiate release during the second stimulation [24,68]. In agreement with these findings, our model establishes the dynamic equilibrium between the RRP (primed pool) and the unprimed pool by ensuring a relatively faster priming rate. As a result, we find here that most synapses operating in the intermediate-release-probability regime, characteristic of hippocampal excitatory synapses, display low STD with PPR < 1 in both WT and disease states (Figure 6B). Strikingly, the FAD-affected synapse displays enhanced presynaptic strength relative to the WT synapse, in contrast to the notion that the activity-dependent tunability of PPR ensures that periods of elevated activity result in a subsequently depressed response. A simple explanation for this is that in the FAD-affected synapse, the elevated residual Ca^2+^ in the nerve terminal after the conditioning stimulus is longer-lasting and facilitates additional release upon subsequent stimulation (Figure 6A). To test the veracity of this claim, we examined whether the $\left[Ca_{AZ}^{2+}\right]$ remains elevated following the second pulse. Consequently, we found that the FAD-associated enhancement of residual $\left[Ca_{AZ}^{2+}\right]$ is sustained after the second stimulation and increases with $P_{r1}$ (Figure 6C). These results suggest that on short timescales, the reduced depression observed in FAD-affected synapses is orchestrated by Ca^2+^ released from internal stores and induces a history-dependent enhancement of STP with respect to the WT synapse.

Although our results suggest that higher-probability synapses consistently express greater depression, it is still unclear whether the probability of transmission of consecutive spikes monotonically relates to the intrinsic release probability, and what role the ER plays in sculpting the concomitant profile. For this purpose, we examine the $P_{r2}$ as a function of $P_{r1}$ (Figure 6D), which reflects the conditional probability that a successful release event on the first pulse is followed by another successful release on the second pulse. Our data reveal that the success of a transmission event in response to the second stimulus depends on that for the first stimulus in a bell-shaped manner, indicating that the probability of vesicle release upon consecutive spikes is attenuated at both low- and high-probability synapses. This implies that synapses with an intermediate number of VGCCs display higher success of transmission of consecutive spikes, in agreement with the idea that the stochasticity/unreliability of transmission probability can enhance the efficacy of information transmission across the synapse [97,98,99]. Unsurprisingly, the FAD-affected synapse facilitates transmission in response to the second stimulus more strongly compared to the WT synapse. This suggests that diseased synapses retain a longer history of Ca^2+^ events, which consequently contributes to the hyperactivation of release at short timescales. To verify this claim, we examined whether the biphasic response of the decay time and cumulative $\left[Ca_{AZ}^{2+}\right]$ due to the first stimulus is sustained after the second stimulus. Indeed, we observe that both decay time and cumulative $\left[Ca_{AZ}^{2+}\right]$ retain their bell-shaped dependence on $Pr_{1}$ (Figure 6E,F). Importantly, the FAD-affected synapse exhibits a markedly enhanced response to the second stimulus following the elevated residual $\left[Ca_{AZ}^{2+}\right]$ due to the first pulse, in agreement with the notion that FAD-affected synapses can result in enhanced excitation of neuronal processes. These findings are consistent with previous work showing that, particularly in the early stages of FAD, over-excitation dominates neuronal circuits with soluble Aβ oligomers and contributes to cognitive dysfunction and impairments [84,100,101]. In summary, FAD-associated enhanced Ca^2+^ release from intracellular stores leads to a history-dependent enhanced STP and hyperactivation of neuronal processes at short timescales with respect to WT synapses. We also notice that the higher coupling strength between the AZ and the microdomain of the IP_3_R cluster exacerbates the enhanced PPR but has a marginal effect on the bell-shaped behavior of $P_{r2}$ as a function of $P_{r1}$ in FAD-affected synapses (Appendix A). 

### 3.5. FAD-Associated Ca^2+^ Rises Differentially Regulate Synchronous and Asynchronous Release during Repetitive Stimulation

Next, we investigate the effect of upregulated cytosolic Ca^2+^ on the synaptic response following trains of stimuli. As in the previous section, we stimulate the synapse with a train of 20 APs delivered at 20 Hz and define facilitation as the ratio of response following the $n^{th}$ stimulus ($R_{n}$) to that of the first ($R_{1}$) averaged over several trials. Therefore, the synaptic response to successive stimuli in the pulse train can either be depressed, with $R_{n}$/$R_{1}$ < 1, or facilitated, with $R_{n}$/$R_{1}$ > 1. In both WT and FAD-affected synapses, repetitive activation leads to the depression of both the peak release rate (Figure 7A) and P_r_ (Figure 7B), which increase with subsequent stimuli. Here, the FAD-affected synapses exhibit a lower peak release rate (Figure 7C) and baseline P_r_ (Figure 7D), leading to lower facilitation that persists throughout activation. The depression following the higher-frequency, longer (~450 ms) stimulus train observed in the FAD-affected synapse results from the rapid depletion of the vesicles in the RRP relative to the WT synapse (Figure 7E).

Although the evoked peak release rate and P_r_ exhibit similar decay as a function of the pulse number, FAD-associated Ca^2+^ disruptions differentially affect these two properties. While the peak release rate in the case of the WT synapse remains mostly higher than that of the FAD-affected synapse, the P_r_ in the case of FAD-affected synapses is consistently higher following successive stimuli (Figure 7C,D). These results indicate that during ongoing activity, ER-mediated Ca^2+^ disruptions drive competition between the primary modes of exocytosis: short-lived synchronous release that dominates evoked release during low-frequency stimulation, and slower asynchronous release that persists for several milliseconds and builds up during higher-frequency stimuli trains [39,69]. Consistent with previous reports, we observe that during the pulse-train depression, synchronous release progressively declines, whereas asynchronous release peaks and subsequently decays with stimulus number (Figure 7G,H). These results also show that while both forms of release compete for the same pool of releasable vesicles, residual Ca^2+^, which builds up during repetitive stimulation, may allow asynchronous release access to a larger subset of the RRP initially [44,45,102]. Our findings here indicate that impairments such as FAD pathology, which trigger elevated levels of residual intracellular [Ca^2+^] (Figure 7F), significantly enhance asynchronous release during the first few pulses (Figure 7G). For both WT and FAD-affected synapses, the decrease in asynchronous release after the peak is dictated by competition with synchronous release for the same vesicle resources, which are rapidly depleted with subsequent stimuli (Figure 7G,H) [45,103]. Interestingly, the greater degree of depression in the FAD-affected synapse is positively correlated with the profound increase and decrease in the rates of asynchronous and synchronous release, respectively. Together, our data suggest that during the stimulus train, AD pathology elicits significantly more asynchronous release at the expense of synchronous release, consistent with the notion that elevated residual Ca^2+^ underlies asynchronous release. Since the majority of evoked exocytosis occurs synchronously with AP-triggered Ca^2+^ influx, the enhanced switch from synchronous to asynchronous release in FAD-affected synapses reflects the increased depression of synaptic transmission with repetitive stimulation. As is clear from Appendix A, the higher coupling between the microdomain around the IP_3_R cluster and AZ exacerbates the synaptic depression in AD-affected synapses more than in WT synapses.

### 3.6. Synchrony of Release Events Is Reduced in FAD-Affected Synapses

Motivated by substantial evidence supporting the reduced temporal coordination of neural activity in AD-affected networks, we next examine the degree of synchronization between stimulus and response in WT and AD-affected synapses during repetitive stimulation [104,105,106]. Given the loose temporal coordination of the neuronal response during asynchronous release, the observed shift from synchronous to asynchronous release during stimulation in FAD should reduce the synchrony between the pulse and response [45]. To test this hypothesis, we measure the event synchrony between each spike in the pulse train and corresponding release using a modified Pinsky–Rinzel algorithm (see Methods for details) [74,75]. In extreme cases, pulse and release event times can either be perfectly aligned, with a synchrony value of 1, reflecting pristine temporal stimulus–response coordination, or desynchronized, where a synchrony value of 0 reflects temporally uncorrelated activity and stimulus patterns.

The phase of individual release events in response to a stimulus train reveals that synapses with FAD pathology exhibit a significant decrease in the coherence of stimulus and release events, an observation that is seen consistently across a wide range of P_r_ values (Figure 8A,B). In silico studies of the effect of presynaptic Ca^2+^ stores on exocytosis at the CA3 terminal suggest that the ER allows highly stochastic hippocampal synapses with low intrinsic release probability to operate with increased reliability [93]. Unsurprisingly, in both the diseased and WT cases, synapses with low P_r_ exhibit relatively higher phase coherence that persists upon subsequent stimulation (Figure 8A,B). Thus, our findings here suggest that the alternative strategy to achieve robust firing rates—employed by synapses with high intrinsic P_r_ that deplete the RRP quickly—may render the synapses unreliable during repetitive stimulation.

Furthermore, a quantitative comparison of the synchrony measure reveals that FAD-affected synapses with altered intracellular Ca^2+^ signaling exhibit significantly reduced temporal coordination of activity as compared to WT synapses. This suggests that in addition to other impairments discussed above, unreliability in the temporal coordination of neuronal activity is likely also a hallmark of FAD-affected synaptic terminals (Figure 8C). This conclusion is consistent with in vivo recordings from neocortical pyramidal neurons, where amyloid-β plaques in APP-Sw (Tg2576 transgenic mice model of AD) increased jitter in the evoked AP, consequently reducing synaptic integration and information transfer [107]. Although our data so far indicate that synapses with higher P_r_ express reduced synchronization in both WT and FAD conditions, the relative sensitivity of synapses with low, intermediate, and high P_r_ to pathological alterations is yet to be explored. Thus, we next examine the relative change in synchrony from WT to FAD condition across a wide range of P_r_ values (Figure 8D). Our results show that there is an inverted bell-shaped relationship between relative synchrony and intrinsic P_r_, suggesting that the physiological (intermediate) P_r_, which allows small hippocampal synapses to operate with reliable firing rates, may also render them more sensitive to pathological alterations. Taken together, these data suggest that the pathophysiological manifestations of impaired intracellular Ca^2+^ handling include suboptimal neuronal synchronization, reducing the fidelity of information integration and transmission. In addition, the results confirm that hippocampal synapses with intermediate P_r_ values exhibit a more severe impairment of their otherwise finely tuned temporal rate codes when subject to FAD-related alterations in Ca^2+^ signaling.

## 4. Discussion

Despite extensive experimental studies reporting the ubiquitous presence of ER in both the axonal and dendritic compartments of neurons, little is known about its role in modifying the major components of synaptic transmission during AD-related pathologies [23,108,109]. In this study, we fill this gap by building a detailed biophysical model that accurately captures the compartmentalized signaling of Ca^2+^ at axonal terminals. In particular, our model incorporates ER-driven and AP-triggered presynaptic Ca^2+^ signaling as well as the resulting release mechanism in WT and FAD-affected synapses. This is especially motivated by reports of distinct regulatory mechanisms for intracellular Ca^2+^, which primarily includes a cluster of ligand-gated IP_3_R Ca^2+^ channels situated on the ER, as well as voltage-activated Ca^2+^ channels constrained to the AZ, which result in tight microdomain signaling [47,53,56,71,110]. In hippocampal synapses, the formation of evoked transient Ca^2+^ microdomains in the AZ is predominantly mediated by the rapid kinetics of P/Q-type VGCCs, which open with minimal delay upon the arrival of AP, while subcellular domains in the vicinity of the ER occur via the stochastic gating of IP_3_Rs that require IP_3_ and Ca^2+^ binding. In order to make meaningful quantitative predictions, we proceeded by developing a model that accurately captures the characteristic IP_3_R’s gating in WT and FAD-affected neurons [18,19,21]. We next incorporated a comprehensive description of presynaptic processes, including evoked Ca^2+^ influx through VGCCs; Ca^2+^ release and uptake by the ER; and synchronous, asynchronous, and spontaneous modes of synaptic transmission. Inspired by several lines of evidence elucidating the existence of a bidirectional interaction of intracellular Ca^2+^ channels and presynaptic VGCCs mediated by stromal interaction molecules (STIM-1) and Orai channels, we have implemented a bidirectional coupling between the ER and AZ in our model that uncovers the unique biphasic dependence of the decay times of release and [Ca^2+^] on the baseline probability [91,111]. Our model incorporates these critical components of presynaptic signaling and especially reproduces the observed spatiotemporal characteristics of intracellular Ca^2+^—rise times, decay times, and amplitudes of corresponding events—and accounts for the stochasticity of presynaptic Ca^2+^ dynamics driven by stochastic channel openings [18,53,67,112]. Release rate, facilitation, and depression vary dramatically among phasic and tonic synapses—phasic synapses are dominated by depression, whereas tonic synapses can be facilitated tremendously by vesicle recruitment [24,68]. Constrained by kinetics data of vesicular release observed in other phasic cell types, our model closely reproduces the essential determinants of neurotransmitter release, which consequently promoted the close estimation of PPR and facilitation requirements necessary to maintain normal plasma membrane function [24,73,94]. Following these independent validations, it is reasonable to assume that the biophysical model and protocols developed in this work are physiologically realistic representations of neuronal processes in control and disease cases.

Broadening of Ca^2+^ waveforms and response profiles affects the reliability of synaptic information transfer at affected terminals. Our results suggest that the FAD-associated increase in Ca^2+^ release from the ER affects nearly all aspects of SVR. Despite the high stochasticity at the hippocampal CA3 terminal, the ER allows individual synapses operating in the low- and intermediate-probability regime to maintain relatively higher reliability of information-rate coding [93]. Here, our results show that the FAD-associated enhanced [Ca^2+^] selectively diminishes the reliability of intermediate-P_r_ synapses, suggesting that low and intermediate P_r_ synapses are more susceptible to FAD-associated Ca^2+^ disruptions. Importantly, the model predicts that the aberrant Ca^2+^ rise in AD-affected neurons may trigger hyperactivity over very short timescales (~30–60 ms) and lowers facilitation during prolonged (~450 ms) stimulation. We also report enhanced excitability in pathological synapses when simulated with higher coupling, which corresponds to a tighter feedback loop between the ER and AZ. Thus, our findings here provide a plausible explanation for why alterations in ER Ca^2+^ handling, which result in excessive efflux, induce a severe perturbation of neuronal processes that can in turn decrease the reliability of information encoded in the firing rate of neurons affected by FAD pathology [93,113]. Overall, our findings provide novel insights into the role of aberrant ER Ca^2+^ release in altering the release profile of a synapse in AD and other neurodegenerative diseases where such Ca^2+^ impairments are observed [114].

Many explorations into the Ca^2+^ dependence of release have proposed that periodic Ca^2+^ release from the ER, which occurs predominantly via the rapid gating kinetics of IP_3_R, could trigger a series of physiological events—such as activating Ca^2+^-sensing G-proteins—that eventually manifest as changes in the global [Ca^2+^] and, in turn, influence spontaneous release: a miniature form of exocytosis [28,29]. Although IP_3_Rs are differentially expressed according to cell type, several studies report the functional involvement of IP_3_Rs in sculpting long-term potentiation (LTP) or depression (LTD) profiles in the CA1 region of the hippocampus [115,116]. In particular, immunocytochemical evidence reveals the expression of IP_3_Rs in presynaptic terminals of the rat CA3-CA1 hippocampal synapses and elucidates their modulatory role in presynaptic neurotransmitter release and synaptic plasticity [109]. Shorter opening times and longer closing times are characteristic features of the gating kinetics of IP_3_Rs in cortical neurons and are essential for the homeostasis of local and global Ca^2+^ in the norm. Consistent with these requirements, studies on AD patients have implicated exaggerated Ca^2+^ release mediated by altered IP_3_R activity in the dysregulation of bulk Ca^2+^, which consequently triggers the progressive loss of synaptic function [117,118,119,120]. From our model, it is clear that these intrinsic biophysical properties of IP_3_R make it highly effective at maintaining physiological bulk Ca^2+^ and P_r_, whereas upregulated IP_3_R orchestrates exaggerated [Ca^2+^] from the ER and in turn increases the P_r_.

Consistent with experimental evidence, our model reproduces the well-established inverse relationship between the P_r_ and PPR but suggests that the enhanced STP over rapid timescales observed in FAD merely indicates hyperactivity rather than increased synaptic reliability [73,121]. The most crucial insight from this finding is that pathological synapses retain a longer history of Ca^2+^ dynamics, which consequently causes them to exhibit enhanced excitation in a paired-pulse protocol. The traditional view claiming the massive reduction in the efficacy of excitatory synaptic transmission in early stages of AD has been challenged recently by several studies reporting aberrant Ca^2+^ homeostasis and hyperactivity in AD-affected neuronal networks [84,100,101]. A key finding of these studies is that hyperactivity is an early dysfunction in hippocampal synapses, whereas neuronal silencing emerges during later stages of the disease. Our model predicts that synaptic facilitation and depression depend on the timescale and frequency of stimulation. Relatively short low-frequency stimuli cause facilitation, which supports the idea that by enhancing potentiation over rapid timescales, aberrant Ca^2+^ release from intracellular stores plays an important role in the history-dependent neuronal hyperactivity observed during early stages of FAD.

Our model shows that long high-frequency pulses trigger depression in FAD-affected synapses, which is governed by the depletion of RRP of vesicles. Experimental evidence reports three distinct molecular pathways for exocytosis [39]. Indeed, synchronous, asynchronous, and spontaneous modes of vesicular release, characterized by distinct release timescales, have been reported in cultured hippocampal synaptic terminals [28,44]. Synaptotagmin-1 (Syt1) and 7 (Syt7) are the Ca^2+^ sensors controlling the timescales of synchronous and asynchronous release in CA3 pyramidal neurons, respectively [39,122]. The rapid kinetics and low Ca^2+^ affinity of Syt1 enable the remarkable temporal precision of synchronous release, where most vesicles immediately fuse with the membrane following stimulation. On the other hand, the slow kinetics of Syt7 promotes the progressive desynchronization of release later in the spike train [123]. Studies on Syt1 knockout mice reported the selective abolishment of synchronous release and an increase in the magnitude of asynchronous release. Likewise, knockdown of Syt7 enhanced synchronous release in zebrafish’s neuromuscular junction, suggesting that the primary modes of exocytosis draw from the same vesicle resources in the RRP, which causes their activity patterns to be negatively correlated [40,44,123,124,125,126]. The most critical insight of these findings is that manipulations that suppress synchronous release increase the vesicle resources available for asynchronous release and indirectly enhance its magnitude. In agreement with the above, during ongoing activity, we observed a shift from synchronous to asynchronous exocytosis, leading to the increased depression of synaptic transmission during FAD pathology.

CA3-CA1 presynaptic terminals are equipped with a conspicuously low release probability that allows them to maintain a delicate balance between facilitation and reliability, giving rise to finely tuned rate codes with remarkable temporal precision. Thus, perturbations of release mechanisms are expected to alter the fidelity of neural rhythms in AD. Indeed, cortical neurons with Aβ peptide expressed reduced N-methyl D-aspartate (NMDA) receptor density, resulting in the rapid and persistent depression of NMDA-evoked currents [106]. Moreover, the severe impairment of evoked synaptic response latency observed in Tg2576 mice overexpressing APP provides direct evidence for the reduced temporal coincidence of response in AD [107]. These findings suggest that impaired response precision is an acute effect of perturbations due to AD that leads to overt cortical deficits. In agreement with these observations, our results reveal the increased latency of release events in FAD and uncover the unique dependence of the synchrony change—from control to FAD—on intrinsic Pr. The loss in temporal coordination of release in FAD is more severely expressed at a physiologically plausible P_r_ range for hippocampal synapses, although lower-P_r_ synapses exhibit relatively elevated temporal precision in both WT and FAD-affected synapses. Thus, despite the high fidelity of hippocampal synapses, their conspicuously low response success may make them more susceptible to AD pathology. We hope that future experiments will uncover the molecular mechanisms underlying the pathological enhancement of susceptibility in low-P_r_ hippocampal synapses.

We remark that while our model is developed to be consistent with most observations in the hippocampal CA3-CA1 synapses, it does not incorporate the uneven distribution of P/Q-type, N-type, and R-type VGCCs specifically found in the AZ of central synapses [66,127]. At hippocampal glutamatergic synapses, Cav2.1—P/Q-type—channels are thought to be most enriched at the presynaptic AZ and predominantly govern Ca^2+^ influx at the axon terminal [52]. Thus, similar to others, we value parsimony and use a formulation with only P/Q VGCCs [67]. Furthermore, results from cultured hippocampal and superior cervical ganglion neurons provide evidence for direct interactions between the release machinery and VGCCs, implying that channel distribution is important for accurately predicting the spatiotemporal profiles of evoked release [128,129]. Our model does not capture the effect of the spatial distribution of VGCCs on synaptic transmission; neither does it incorporate other mechanisms for paired-pulse modulation expressed at putative single hippocampal synapses—lateral inhibition and release inactivation [73,94]. Thus, despite reproducing the observed inverse relationship between paired-pulse facilitation and initial P_r_, our model falls short of the measured values, suggesting that we cannot eliminate additional mechanisms when investigating the interplay between residual Ca^2+^, facilitation, and depression. Although IP_3_R-dependent modulation of cytosolic Ca^2+^ is usually adequate for explaining the critical aspects of ER Ca^2+^ release and regulation of neurotransmission, the upregulation of RyR expression and modulation of IP_3_Rs’ gating due to Ca^2+^ release through RyRs (and vice versa) are also thought to play a key role in the aberrant Ca^2+^ release from the ER, as well as the propagation of presynaptic signals [10,11,12,13,14,15,16,17,18,19,20,21,22,130]. It has been suggested that at the hippocampal Schaffer-collateral pathway, presynaptic presenilin inactivation perturbs STP and facilitation via impaired RyR function [36,131]. Furthermore, in 3xTg-AD mice, deviant RyR activity triggers Ca^2+^ signaling alterations that promote synaptic depression [37]. However, our model does not describe the contribution of presynaptic RyRs to vesicular release, as the biophysical properties of these receptors yield a distinct temporal range of Ca^2+^ transients that can modulate LTP/LTD [132,133]. Presenilin mutations account for a significant portion of FAD cases; however, our current model does not incorporate ER-resident Ca^2+^ permeable leak channels formed by Presenilins that are disrupted in FAD-affected neurons [134]. Another key factor missing from our model is synaptic mitochondria. Mitochondria play a key role in shaping Ca^2+^ gradients in synaptic terminals. Synaptic mitochondria are among the earliest targets in AD. Among other aspects, the ATP production and Ca^2+^ buffering capacities of mitochondria are severely disrupted [135,136]. All these issues will be the subjects of our future research.

In summary, we have leveraged diverse experimental data to model Ca^2+^ homeostasis in the axonal terminal and explore how exocytosis is affected in FAD. Motivated by the difficulty in probing signaling cascades at the AZ of small hippocampal synapses, our main goal was to build a comprehensive but simple framework for unravelling the role of enhanced Ca^2+^ release from the ER in SVR during FAD pathology. In addition to agreeing closely with several observations about the kinetics of IP_3_Rs, SVR, and synaptic plasticity in both WT and diseased synapses, our modelling work provides key insights into impaired presynaptic function in FAD. Specifically, we make five key predictions: The overall P_r_ in response to a single AP is upregulated in FAD-affected synapses;Short-lived low-frequency stimuli promote potentiation in FAD-affected synapses;During sustained high-frequency stimulation, FAD-affected terminals exhibit enhanced depression;FAD-affected synapses operate less reliably, andThe effect of FAD pathology is exacerbated in synapses with low to intermediate P_r_. 

Although extensive experimental evidence corroborates our findings and demonstrates the significance of ER Ca^2+^ dysregulation and loss of synaptic integrity in FAD, SVR therapeutics remain underdeveloped. For example, the current lack of validated in vivo biomarkers that can be used as proxies of synaptic dysfunction has directed most translational efforts towards indirect measures of synaptic integrity in AD [137]. Our findings suggest that targeting the essential components of neurotransmission as well as related pathways of the ER could potentially aid biomarker identification and drug discovery efforts in FAD. In future research, we expect to incorporate a better and comprehensive understanding of the mechanisms underlying neuronal Ca^2+^ dysregulation to provide better insights into the disease pathogenesis and therapeutic directions. Taken together with the aforementioned limitations, our analysis highlights the need for further studies investigating the role of perturbed Ca^2+^ signaling due to intracellular organelles such as the ER and mitochondria in cognitive deficits associated with AD and other neurodegenerative diseases.

## Figures and Tables

**Figure 1 cells-11-02167-f001:**
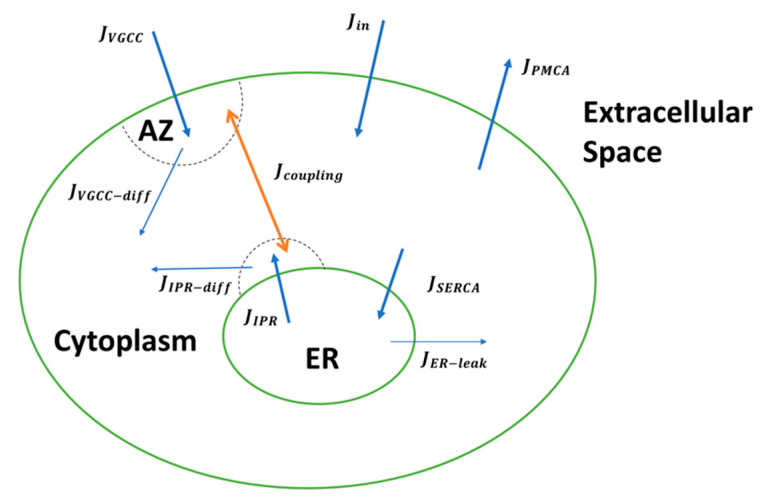
Schematic of the overall multi-compartmental Ca^2+^ model. The arrowheads show the direction of the fluxes involved and the dotted half circles signify the Ca^2+^ domains around the IP_3_Rs and VGCC clusters.

**Figure 4 cells-11-02167-f004:**
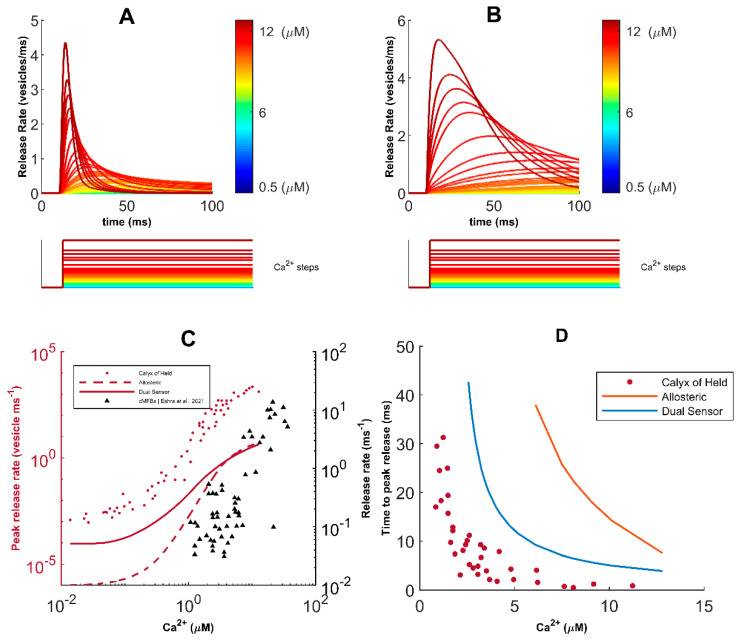
Characterization of neurotransmission in response to [$Ca_{cyt}^{2+}$] steps: (**A**) Total release events obtained from a single dual−sensor fusion process after clamping [$Ca_{cyt}^{2+}$ ] at different values; (**B**) Release profile following allosteric fusion in response to stepwise [$Ca_{cyt}^{2+}$ ] clamp; (**C**) Regulation of the peak release rate in response to clamped [$Ca_{cyt}^{2+}$ ] levels shows lower and right-shifted dose responses relative to the experimental data for the Calyx of Held; (**D**) [$Ca_{cyt}^{2+}$ ] dependence of time−to−peak rate indicates exponentially decreasing but longer time delay to peak release when matched with data for the Calyx of Held. Experimental values shown for comparison in (**C**,**D**) are from [72].

**Figure 5 cells-11-02167-f005:**
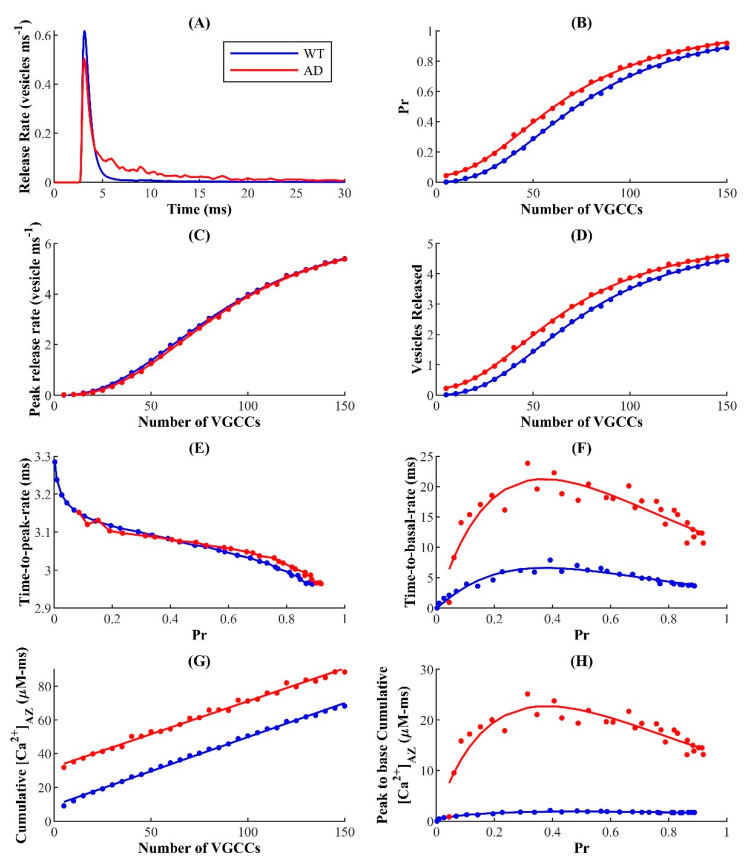
ER-driven upregulation of cytosolic Ca^2+^ leads to enhanced synaptic vesicle release in FAD: (**A**) Neurotransmitter release rate in response to a single AP in WT and AD−affected synapse. Change in the P_r_ of a single synaptic vesicle (**B**); peak release rate (**C**); and the average number of vesicles released (**D**) as functions of the number of VGCCs; (**E**) Time delay of peak release rate and (**F**) decay time to basal release rate as functions of P_r_; (**G**) Change in $\left[Ca_{AZ}^{2+}\right]$ with number of VGCCs; (**H**) Cumulative Ca^2+^ from peak to basal level as a function of P_r_.

**Figure 6 cells-11-02167-f006:**
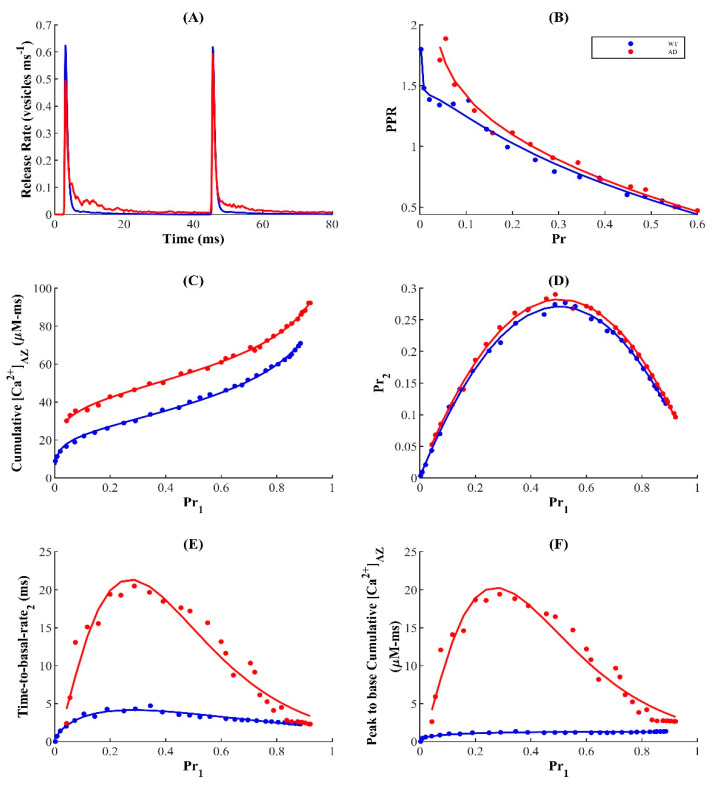
FAD−associated Ca^2+^ upregulation enhances STF: (**A**) Release profile following paired−pulse stimulation protocol; (**B**) PPR is inversely related to intrinsic P_r_ (obtained after first pulse), and is higher in the AD−affected synapse; (**C**) Similar to the first pulse (Figure 3G), cumulative $\left[Ca_{AZ}^{2+}\right]$ after the second pulse increases with the P_r_ and is higher in the AD−affected synapse; (**D**) Pr in response to the second pulse ($P_{r2}$ as a function of P_r_ following the first pulse ($P_{r1}$ Higher values indicate that the synapse responds more strongly to the subsequent stimulus in a paired−pulse protocol; (**E**) Decay time of release rate after second pulse also exhibits a biphasic behavior; (**F**) Cumulative $\left[Ca_{AZ}^{2+}\right]$ following the second pulse reflects the biphasic behavior observed in time delay of peak−to−basal release rate in panel (**E**).

**Figure 7 cells-11-02167-f007:**
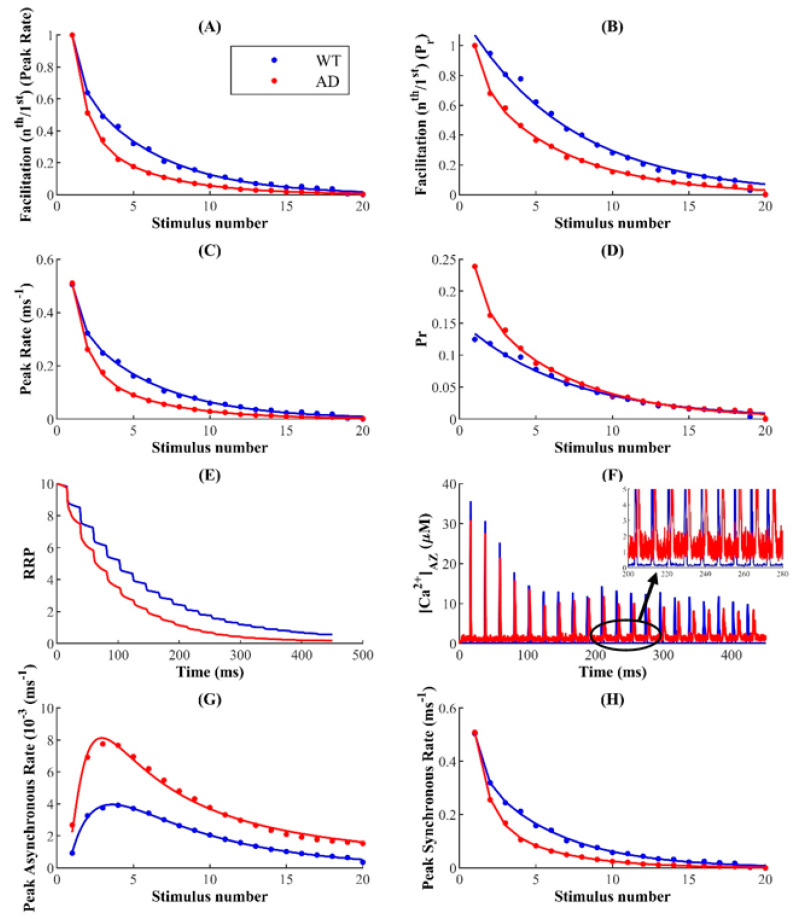
The AD−affected synapse exhibits stronger depression in response to 20 pulse stimulus train delivered at 20 Hz: Facilitation obtained from peak rate (**A**) and P_r_ (**B**) shows that AD pathology induces more severe depression relative to control conditions. Peak release rate (**C**) and P_r_ (**D**) following each AP in the train. (**E**) Pulse train depression is primarily governed by RRP depletion, which is more severe in synapses with AD pathology. (**F**) ${\left[Ca^{2+}\right]}_{AZ}$ (top) and zoom-in (inset) showing the differences in basal ${\left[Ca^{2+}\right]}_{AZ}$ levels. (**G**) Asynchronous release peaks and subsequently decays following depletion of RRP. (**H**) Peak synchronous release mimics the response seen in the overall release.

**Figure 8 cells-11-02167-f008:**
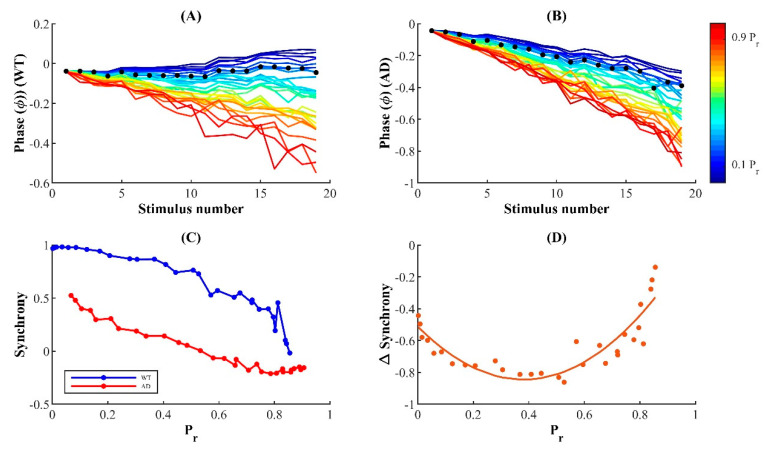
AD−related Ca^2+^ disruptions impair spike–SVR synchrony: The phase of individual release events with respect to AP in WT (**A**) and AD−affected (**B**) synapses over a wide range of P_r_ values. Black dots indicate the event phases corresponding to 35 VGCCs with initial P_r_ during pulse train (P_r1_) equal to 0.14; (**C**) Synchrony of release event in response to the preceding pulse in WT and AD−affected synapses; (**D**) The magnitude of relative synchrony change from WT to AD conditions as a function of initial release probability during prolonged stimulation.

**Table 1 cells-11-02167-t001:** Ca^2+^ fluxes in the model.

Flux (Reference)	Equation
Basal leak and R/SOCC [49,50]	$J_{in}\;$ $=J_{leakin}$ $+V_{leakin}IP_{3}$
Ca^2+^ diffusion from the IP_3_R cluster [47]	$J_{IPR-diff}=k_{IPR-diff}\left(Ca_{IPR_{n}}^{2+}-Ca_{cyt}^{2+}\right)$
PMCA [47]	$J_{PMCA}=V_{PMCA}\frac{{\left(Ca_{cyt}^{2+}\right)}^{n}}{{\left(Ca_{cyt}^{2+}\right)}^{n}+K_{PMCA}{}^{n}}$
SERCA [47]	$J_{SERCA}=V_{SERCA}\frac{{\left(Ca_{cyt}^{2+}\right)}^{n}}{{\left(Ca_{cyt}^{2+}\right)}^{n}+K_{SERCA}{}^{n}}$
ER leak [47]	$J_{ER-leak}=k_{ER-leak}\left(Ca_{ER}^{2+}-Ca_{cyt}^{2+}\right)$
IP_3_ receptor [47]	$J_{IPR}=k_{IPR}P_{o}\left(Ca_{ER}^{2+}-Ca_{IPR_{n}}^{2+}\right)$ $P_{o}=N_{open}/N_{IP_{3}R}$
Ca^2+^ diffusion from the VGCC cluster	$J_{VGCC-diff}=k_{VGCC-diff}\left(Ca_{AZ}^{2+}-Ca_{cyt}^{2+}\right)$
ER–AZ coupling	$J_{coupling}=V_{c}\frac{\left(Ca_{AZ}^{2+}{}^{2}-\bar{k}\left(Ca_{IP_{3}R_{n}}^{2+}{}^{2}\right)\right)}{Ca_{AZ}^{2+}{}^{2}-K_{c}{}^{2}}$
VGCC [53]	$J_{VGCC}=\frac{-I_{Ca^{2+}}}{zFV_{cell}}$ $I_{Ca^{2+}}=channel\;density\;\cdot cluster\;area\cdot I_{s}$ $I_{s}=gP_{o}\left(V-E_{Ca^{2+}}\right),\;$ $P_{o}=N_{open}/N_{VGCC}$ $channel\;density=\frac{N_{VGCC}}{\left(AZ_{area}\cdot N_{AZ}\right)}$

**Table 2 cells-11-02167-t002:** Parameter values for the Ca^2+^ dynamics.

Category	Parameters	Description	Value/Units	Notes
Cellular	$J_{leakin}$	Plasma membrane leak influx	0.03115 μM·ms^−1^	From [47]
	$V_{leakin}$	R/SOCC flux coefficient	0.2 ms^−1^	From [47]
	$k_{IPR-diff}$	$Ca_{IP_{3}R_{n}}^{2+}$ diffusional flux coefficient	10 ms^−1^	From [47]
	$k_{IPR}$	IP_3_R flux coefficient	5 ms^−1^	Modified from [47]
	$N_{IPR}$	Total number of IP_3_R channels	10	
	$k_{ER-leak}$	ER leak flux coefficient	0.0022 ms^−1^	Modified from [47]
PMCA	$V_{PMCA}$	Maximum capacity of PMCA	3.195 μM·ms^−1^	Modified from [47]
	$K_{PMCA}$	Half activation PMCA constant	0.5 μM	From [47]
	$n_{P}$	Hill coefficient of PMCA	2	From [47]
SERCA	$V_{SERCA}$	Maximum capacity of SERCA	10 μM·ms^−1^	From [47]
	$K_{SERCA}$	Half maximal activation SERCA	0.26 μM	From [47]
	$n_{S}$	Hill coefficient of SERCA	1.75	From [47]
VGCC and Coupling	$k_{VGCC-diff}$	$Ca_{AZ}^{2+}$ diffusional flux coefficient	0.071 ms^−1^	
	$V_{cell}$	Terminal volume	1.22 × 10^−6^ L	Modified from [53]
	cluster area	Area of VGCC cluster	0.001963 μm^2^	Modified from [53]
	$AZ_{area}$	Active zone area	0.04 μm^2^	Modified from [51,53]
	$N_{AZ}$	Active zone number	1.3	Modified from [53]
	$V_{c}$	Maximum capacity of transfer component	118 μM·ms^−1^	
	$\bar{k}$	Concentrating power of the transfer components	High coupling: 15 μM	
			Low coupling: 5 μM	
	$K_{c}$	Half-maximal transfer rate	High coupling: 10 μM	
			Low coupling: 20 μM	

**Table 3 cells-11-02167-t003:** Additional parameters used in the Ca^2+^ dynamics.

Parameter	Value/Units	Notes
$\mathrm{Resting}\;\mathrm{cytosol}\;[Ca_{cyt}^{2+}$]	0.1 μM	[58]
$\mathrm{Resting}\;\mathrm{AZ}\;[Ca_{AZ}^{2+}$]	0.05 μM	[53]
$\mathrm{Resting}\;\mathrm{total}\;[Ca_{total}^{2+}$]	56 μM	
$\mathrm{Resting}\;\left[IP_{3}\right]$	0.1 μM	[58]
$\mathrm{Extracellular}\;[Ca_{ext}^{2+}$]	2.0 mM	[53]
$\delta_{1}$	100	Cytoplasmic to ER microdomain volume ratio [47]
$\delta_{2}$	10	Cytoplasmic to ER volume ratio [47]
$\delta_{3}$	60	Cytoplasmic to VGCC microdomain volume ratio

**Table 4 cells-11-02167-t004:** IP_3_ fluxes in the model.

Flux (Reference)	Equation
PLC_δ_ [57,58]	$J_{PLC}\;$ $={\bar{V}}_{PLC}\frac{{\left(Ca_{cyt}^{2+}\right)}^{2}}{{\left(Ca_{cyt}^{2+}\right)}^{2}+K_{PLC}{}^{2}}$ ${\bar{V}}_{PLC}=V_{PLC}PLC$ $V_{PLC}=V_{0}+V_{Q}\frac{q^{2}}{q^{2}+K_{Q}{}^{2}}$ $q=H\left(t-t_{1}\right)*A\beta *e^{-r\left(t-t_{1}\right)H\left(t-t_{1}\right)}$ $\rho =V_{R}\frac{q}{q+K_{R}}$ $\frac{d}{dt}PLC=k_{f_{PLC}}G\left(PLC_{tot}-PLC\right)-k_{b_{PLC}}PLC$ $\frac{d}{dt}G=k_{f_{G}}\left(\rho +\delta \right)\left(G_{tot}-G\right)-k_{b_{G}}G$
IP_3_ degradation [57]	$J_{\deg}$ $=\left(\eta \frac{{\left(Ca_{cyt}^{2+}\right)}^{2}}{{\left(Ca_{cyt}^{2+}\right)}^{2}+K_{IP_{3}k}{}^{2}}+\left(1-\eta \right)\right)IP_{3}$ $\eta =k_{3k}/\left(k_{3k}+k_{5p}\right)$
Time constant of IP_3_ turnover	$\tau_{IP_{3}}=1/\left(k_{3k}+k_{5p}\right)$

**Table 5 cells-11-02167-t005:** Transition rates used in the IP_3_R model.

Rate/Parameter	Equation
Transition rates	$k_{RA}={\left[1\times \left(\frac{1}{j_{01}{\left[Ca^{2+}\right]}_{IPR_{n}}}+\frac{1}{j_{12}{\left({\left[Ca^{2+}\right]}_{IPR_{n}}\right)}^{2}}\right)\right]}^{-1}$ $k_{AR}={\left[K_{A}{\left({\left[Ca^{2+}\right]}_{IPR_{n}}\right)}^{2}\times \left(\frac{1}{j_{01}{\left[Ca^{2+}\right]}_{IPR_{n}}}+\frac{1}{j_{12}{\left({\left[Ca^{2+}\right]}_{IPR_{n}}\right)}^{2}}\right)\right]}^{-1}$ $k_{AO}={\left[K_{A}{\left({\left[Ca^{2+}\right]}_{IPR_{n}}\right)}^{2}\times \left(\frac{1}{j_{22}{\left({\left[Ca^{2+}\right]}_{IPR_{n}}\right)}^{2}}\right)\right]}^{-1}$ $k_{OA}={\left[K_{O}{\left({\left[Ca^{2+}\right]}_{IPR_{n}}\right)}^{2}\times \left(\frac{1}{j_{22}{\left({\left[Ca^{2+}\right]}_{IPR_{n}}\right)}^{2}}\right)\right]}^{-1}$ $k_{OI}={\left[K_{O}{\left({\left[Ca^{2+}\right]}_{IPR_{n}}\right)}^{2}\times \left(\frac{1}{j_{23}{\left({\left[Ca^{2+}\right]}_{IPR_{n}}\right)}^{3}}+\frac{1}{j_{45}{\left({\left[Ca^{2+}\right]}_{IPR_{n}}\right)}^{5}}\right)\right]}^{-1}$ $k_{IO}={\left[K_{I}{\left({\left[Ca^{2+}\right]}_{IPR_{n}}\right)}^{5}\times \left(\frac{1}{j_{23}{\left({\left[Ca^{2+}\right]}_{IPR_{n}}\right)}^{3}}+\frac{1}{j_{45}{\left({\left[Ca^{2+}\right]}_{IPR_{n}}\right)}^{5}}\right)\right]}^{-1}$ $k_{RI}={\left[1\times \left(\frac{1}{{\tilde{j}}_{01}{\left[Ca^{2+}\right]}_{IPR_{n}}}+\frac{1}{{\tilde{j}}_{45}{\left({\left[Ca^{2+}\right]}_{IPR_{n}}\right)}^{5}}\right)\right]}^{-1}$ $k_{IR}={\left[K_{I}{\left({\left[Ca^{2+}\right]}_{IPR_{n}}\right)}^{5}\times \left(\frac{1}{{\tilde{j}}_{01}{\left[Ca^{2+}\right]}_{IPR_{n}}}+\frac{1}{{\tilde{j}}_{45}{\left({\left[Ca^{2+}\right]}_{IPR_{n}}\right)}^{5}}\right)\right]}^{-1}$
Occupancy parameters	$K_{O}=a_{1}\frac{{\left[IP_{3}\right]}^{n_{O}}}{{\left[IP_{3}\right]}^{n_{O}}+K_{Od}{}^{n_{O}}}$ $K_{A}=a_{2}\frac{{\left[IP_{3}\right]}^{n_{A}}}{{\left[IP_{3}\right]}^{n_{A}}+K_{Ad}{}^{n_{A}}}$ $K_{I}=a_{2}\frac{{\left[IP_{3}\right]}^{n_{I}}}{{\left[IP_{3}\right]}^{n_{I}}+K_{Id}{}^{n_{I}}}$

**Table 6 cells-11-02167-t006:** Parameter values for the IP_3_ dynamics and IP_3_R.

Category/References	Parameters	Description	WT	AD
IP_3_ Model (From [58])	$V_{0}$	PLC-mediated IP_3_ production	0.15 μM	0.19 μM
	$V_{Q}$	Control parameter for influence of Aβ on IP_3_	7.82 μM	380 μM
	$K_{Q}$	PLC dissociation constant	0.0086 μg/mL	0.0086 μg/mL
	$K_{IP_{3}k}$	Half-activation for 3-kinase	0.6 μM	1.6 μM
	$K_{PLC}$	PLC sensitivity to Ca^2+^	0.01 μM	0.016 μM
	$k_{3k}$	IP_3_ phosphorylation rate	1.5 μs^−1^	0.7 μs^−1^
	$k_{5p}$	IP_3_ dephosphorylation rate	0.01 μs^−1^	0.005 μs^−1^
PLC (Modified from [58])	$k_{f_{PLC}}$	PLC-protein activation rate	0.35 μs^−1^	0.75 μs^−1^
	$k_{b_{PLC}}$	PLC-protein deactivation rate	22 μs^−1^	200 μs^−1^
	$PLC_{tot}$	Scaled total number of PLC	1	1
G-Protein (From [58])	$k_{f_{G}}$	G-protein activation rate	0.33 μs^−1^	0.047 μs^−1^
	$k_{b_{G}}$	G-protein deactivation rate	2.17 μs^−1^	4.7 μs^−1^
	δ	G-protein intrinsic activity	0.01	0.012
	$V_{R}$	Maximal G-protein activation	7.4	10
	$K_{R}$	Aβ producing half-activation	4467 μg/mL	2000 μg/mL
	$G_{tot}$	Scaled total number of G-protein	1	1
IP_3_R (Fit to Experiment)	$a_{1}$		17.05043 μM^−2^	1.108278 × 10^2^ μM^−2^
	$n_{O}$		2.473407	2.473407
	$K_{Od}$		0.909078 μM	0.909078 μM
	$a_{2}$		18.49186 μM^−2^	18.49186 μM^−2^
	$n_{A}$		0.093452	0.093452
	$K_{Ad}$		1.955650 μM	1.955650 μM
	$a_{3}$		2.340259 × 10^2^ μM^−5^	1.4041556 × 10^2^ μM^−5^
	$n_{I}$		56.84823	56.84823
	$K_{Id}$		0.089938 μM	0.089938 μM
	$j_{01}$		3.031635 × 10^2^ μM^−1^ ms^−1^	3.031635 × 10^2^ μM^−1^ ms^−1^
	$j_{12}$		3.230063 × 10^2^ μM^−2^ ms^−1^	3.230063 × 10^2^ μM^−2^ ms^−1^
	$j_{22}$		4.814111 μM^−2^ ms^−1^	5.3978052 μM^−2^ ms^−1^
	$j_{23}$		5.356155 μM^−3^ ms^−1^	2.0652269 × 10^3^ μM^−3^ ms^−1^
	$j_{45}$		5.625616 μM^−5^ ms^−1^	5.4319289 μM^−5^ ms^−1^
	${\tilde{j}}_{01}$		3.013284 × 10^2^ μM^−1^ ms^−1^	3.013284 × 10^2^ μM^−1^ ms^−1^
	${\tilde{j}}_{45}$		2.648741 μM^−5^ ms^−1^	8.512829 × 10^−8^ μM^−5^ ms^−1^

## Data Availability

The complete model code as well as analysis scripts will be posted on our lab’s webpage after the manuscript is accepted for publication.

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
