# Peer review of "Upregulated Ca^2+^ Release from the Endoplasmic Reticulum Leads to Impaired Presynaptic Function in Familial Alzheimer’s Disease"

_cells, 2022, doi:10.3390/cells11142167_

Round 1

Reviewer 1 Report

Dr Ghanim Ullah and colleagues report on their modelling of intracellular calcium signaling and neurotransmitter release in the active zone of hippocampal neurons to investigate the effects of abberant calcium signaling in Alzheimer´s disease.

From their reported work, the authors conclude that enhanced calcium release from the ER increases the probability of neurotransmitter release in Alzheimer´s disease and that this and their additional findings support a crucial role of altered calcium homeostasis as mediated by intracellular stores in Alzheimer´s disease.

Undoubtedly, the manuscript deals with a formidable medical problem. It is written very well and describes the limitations of the approach in the Discussion to a reasonable extent.

However, what I find missing is the idea, best given in the introduction, where this approach is supposed to eventually lead, i.e. after all current limitations have been integrated into the model. Furthermore, it would be interesting to learn early in the manuscript how realistic the authors see their chances to reach their goal.

A second problem I have with the manuscript is that it ascribes the abberant calcium signaling in Alzheimer´s disease to the effect of mutated presenilin on IP3Rs (lines 314-316). However, even in familial Alzheimer´s disease this is not true for all cases. Furthermore, the authors are neglecting the seminal paper that characterized presinilin itself as a significant ER calcium leak channel (Tu H, Nelson O, Bezprozvanny A, Wang Z, Lee S-F, Hao Y-H, Serneels L, De Strooper B, Yu G, Bezprozvanny I. Presenilins form ER Ca2+ leak channels, a function disrupted by familial Alzheimer´s disease-linked mutations. Cell 2006; 126:981-93.) Thus the starting assumption is not generally true. That needs to be dealt with. Furthermore, it may be safer to limit the model, i.e. title and various statements, to familial Alzheimer´s disease, where presenilin mutant variants indeed play an important role.

Minor points:

The title and many other statements give the impression that the presented work explains it all what we ever wanted to know about Alzheimer´s disease. Obvioulsy, I am exaggerating here, but nevertheless feel that the title and many statements should be toned down a bit to avoid my impression by the reader.

Another aspect to be considered by the editors is whether or not this manuscript is really suited for Cells. I remain unconvinced.

Reviewer 2 Report

In this manuscript, Temitope et al investigate the function of unregulated calcium releases from the ER in neurotransmitter release at presynaptic terminals in Alzheimer's disease. The authors employed an interesting method by combining  the IP3R model, membrane voltage dynamics, and voltage-gated calcium channel with release model, and was able to show that unregulated calcium from ER induces enhanced short-term plasticity and unreliable synaptic functions. Although the manuscript does not cover the full map of calcium dynamic at presynaptic terminal, such as calcium released from mitochondira, it provides an novel insight for understanding the dynamic calcium in neurotransmitter release and its effect in AD. I am happy to support the manuscript to be published in Cells.

Reviewer 3 Report

The manuscript of Adeoye et. al. reports on a comprehensive in-silico study of the role of internal calcium store in dysregulation of synaptic transmission in Alzheimer's disease (AD). The authors presented the mathematical model that takes into account most of the main players of calcium homeostasis in neurons and synaptic vesicles release. This model is in agreement with previously reported experimental data and mimics very well the stimulation of IP3 receptors of the endoplasmic reticulum (ER) in physiological conditions and in AD. The model also mimics closely the synaptic release behavior in Calyx of Held. The authors report very interesting predictions such as the increase of probability of neurotransmitter release in AD stimulated by the calcium from internal stores and that impact of ER on synaptic transmission depends on the duration of calcium stimulus. Overall, the study is well designed and the manuscript is logically structured. The conclusions are supported by the prediction and are of great interest to the broad scientific readership.

I have only one major comment and several minor concerns.

The major comment is that model does not take into account mitochondria which are also considered to be internal calcium stores, but the authors do not discuss mitochondria in the introduction part and do not incorporate mitochondrial Ca2+ uptake and release in their calcium model. Could the authors please elaborate on this part in their manuscript?

Minor concerns:

It would be helpful to add the scheme of calcium homeostasis in the introduction part of the manuscript. The comparison of such a scheme with figure 1 of the manuscript will give the reader an idea of how accurate the modeling is.

Could the authors please elaborate on the limitations of the study in the discussion part?

Not all the tables with values of parameters of the equations have references.

Reviewer 4 Report

Cells; MS# cells-1736094

Title: Upregulated Ca2+ release from the endoplasmic reticulum leads to impaired presynaptic function in Alzheimer’s disease

Adeoye and co-authors conducted an in silico study how protein mutations of Alzheimer’s disease (e.g., presenilin) impact coupling between presynaptic calcium and exocytosis of neurotransmitters in the hippocampus. The authors assembled experimental data throughout the literature and created a model to approximate calcium homeostasis with an emphasis on microdomain signaling among the “active zone” (voltage-gated calcium influx) and the endoplasmic reticulum (intracellular calcium sequestration and release). As a result of their findings, the authors generally conclude that conditions of AD underlie a pathological upregulation of calcium release from the ER that, in turn, compromises reliable neurotransmission. I only have one minor comment below for improving the manuscript.

(1) The manuscript needs to be checked thoroughly for typos (e.g., Figure 2d, “channel attachement”; Line 522, “…of the of…”) and incorrect citations (e.g., Line 204, no 3xTg AD animals in ref 20; Line 321, data referring to reference 21 and not 76).

Reviewer 5 Report

I read with great interest the Manuscript titled “Upregulated Ca2+ release from the endoplasmic reticulum leads to impaired presynaptic function in Alzheimer’s disease which falls within the aim of Cells. In my honest opinion, the topic is interesting enough to attract the readers' attention.

However, I suggest authors to consider the following recommendations:

In the introduction authors states: AD manifests as progressive memory impairment initially and faster rate of cognitive decline and neurodegeneration in later stages. Despite the convoluted etiology of AD, experimental and theoretical investigation suggests that synapses are the primary targets in the early stage of the disease [3, 4]. Histologically, the AD brain is marked by extracellular deposition of senile beta-amyloid (Aβ) plaques—the result of abnormalities in the genes encoding amyloid precursor protein (APP) or intramembrane protease presenilin 1 and 2 (PS1, PS2)”.

An increased line of evidence indicates that AD can be considered also characterized by non-cognitive symptoms (i.e., DOI: 10.3389/fneur.2022.832199). I suggest authors to add this information.

AD can be classified as sporadic or genetic and several hypotheses has been formulated to explain the pathogenesis (https://doi.org/10.3390/antibiotics11060726). Thus, it is not correct to states that Ab plaques are only the results of abnormalities in the genes encoding for APP, PS1 and PS2, although the heritability has been estimate between 58–79% and over 90% for late-onset and early-onset AD, respectively (DOI: 10.1038/s41593-020-0599-5). I suggest authors to add this information and to include this evidence in the interpretation and generalizability of their results (and, if appropriate, in the limitations section).

I wish you best of luck with your research!

Round 2

Reviewer 1 Report

well done